https://doi.org/10.1038/s42003-020-0942-0　**OPEN**

# Methylation deficiency disrupts biological rhythms from bacteria to humans

Jean-Michel Fustin [1,16,18 ✉], Shiqi Ye[1,18], Christin Rakers [2], Kensuke Kaneko[3], Kazuki Fukumoto[1], Mayu Yamano[1], Marijke Versteven[4], Ellen Grünewald[5], Samantha J. Cargill[5], T. Katherine Tamai [6], Yao Xu[7], Maria Luísa Jabbur[7], Rika Kojima[8], Melisa L. Lamberti[9], Kumiko Yoshioka-Kobayashi[10], David Whitmore [11], Stephanie Tammam[12], P. Lynne Howell [12,13], Ryoichiro Kageyama[10], Takuya Matsuo[14], Ralf Stanewsky [4], Diego A. Golombek[9], Carl Hirschie Johnson[7], Hideaki Kakeya [3], Gerben van Ooijen [5] & Hitoshi Okamura[15,17 ✉]

The methyl cycle is a universal metabolic pathway providing methyl groups for the methylation of nuclei acids and proteins, regulating all aspects of cellular physiology. We have previously shown that methyl cycle inhibition in mammals strongly affects circadian rhythms. Since the methyl cycle and circadian clocks have evolved early during evolution and operate in organisms across the tree of life, we sought to determine whether the link between the two is also conserved. Here, we show that methyl cycle inhibition affects biological rhythms in species ranging from unicellular algae to humans, separated by more than 1 billion years of evolution. In contrast, the cyanobacterial clock is resistant to methyl cycle inhibition, although we demonstrate that methylations themselves regulate circadian rhythms in this organism. Mammalian cells with a rewired bacteria-like methyl cycle are protected, like cyanobacteria, from methyl cycle inhibition, providing interesting new possibilities for the treatment of methylation deficiencies.

[1] Graduate School of Pharmaceutical Sciences, Laboratory of Molecular Metabology, Kyoto University, Kyoto, Japan. [2] Graduate School of Pharmaceutical Sciences, Kyoto University, Kyoto, Japan. [3] Graduate School of Pharmaceutical Sciences, Department of System Chemotherapy and Molecular Sciences, Kyoto University, Kyoto, Japan. [4] Institute of Neuro- and Behavioral Biology, University of Münster, Münster, Germany. [5] School of Biological Sciences, University of Edinburgh, Edinburgh, UK. [6] Department of Psychiatry and Biobehavioral Sciences, University of California, Los Angeles, Los Angeles, CA, USA. [7] Department of Biological Sciences, Vanderbilt University, Nashville, TN, USA. [8] Karolinska Institute, Stockholm, Sweden. [9] Department of Science and Technology, National University of Quilmes/CONICET, Buenos Aires, Argentina. [10] Institute for Frontier Life and Medical Sciences, Kyoto University, Kyoto, Japan. [11] Centre for Cell and Molecular Dynamics, Department of Cell and Developmental Biology, University College London, London, UK. [12] Molecular Medicine, Peter Gilgan Centre for Research and Learning (PGCRL), The Hospital for Sick Children, Toronto, ON, Canada. [13] Department of Biochemistry, University of Toronto, Toronto, ON, Canada. [14] Center for Gene Research, Nagoya University, Nagoya, Japan. [15] Graduate School of Pharmaceutical Sciences, Laboratory of Molecular Brain Science, Kyoto University, Kyoto, Japan. [16] Present address: The University of Manchester, Faculty of Biology, Medicine and Health, Oxford Road, Manchester M13 9PL, UK. [17] Present address: Kyoto University, Graduate School of Medicine, Department of Neuroscience, Division of Physiology and Neurobiology, Yoshida-Konoe-cho, Oxford Road, Sakyo-ku, Kyoto 606-8501, Japan. [18] These authors contributed equally: Jean-Michel Fustin, Shiqi Ye. ✉email: jean-michel.fustin@manchester.ac.uk; okamura.hitoshi.4u@kyoto-u.ac.jp

Methylation reactions start with the metabolization of methionine into S-adenosylmethionine, or SAM: the universal methyl donor co-substrate in the trans-methylation of nucleic acids, proteins, carbohydrates, phospholipids and small molecules. During the methylation process, SAM is converted into adenosylhomocysteine (SAH) that is rapidly hydrolyzed into homocysteine (Hcy) and adenosine by the enzyme *adenosylhomocysteinase* (AHCY) to prevent competitive inhibition of methyltransferase enzymes by SAH. The ratio SAM/SAH is critical and is a measure of the methylation potential: the tendency to methylate biomolecules[1–3]. Methylation deficiencies, either from poor diet or genetic polymorphisms, contribute to the etiology of many pathologies: cancer, atherosclerosis, birth defects, aging, diabetes and pancreatic toxicity, hepatotoxicity and neurological disturbances[4].

An endogenous circadian clock evolved to anticipate the daily cycles of light and darkness has been found in many organisms, from cyanobacteria to humans. Transcription-translation feedback loops of "clock genes" directly or indirectly regulating their own transcription underlie many functions of the clock, and drive oscillations of output genes controlling physiology and behavior. Some molecular components of the clock are remarkably conserved in Metazoa, notably the genes *Clock* and *Period*, coding for transcription factors, with *Clock* activating the transcription of *Per* and *Per* inhibiting its own transcription[5]. In 2013, we reported that inhibition of the methyl cycle by AHCY inhibitors strongly affected the circadian clock in mouse and human cells[6]. We now show that the link between methylation and the circadian clock we uncovered in mammals has been conserved during more than 2.5 billion years of evolution, and that circadian rhythm perturbations can be used as a quantitative gauge for the physiological consequences of methylation deficiency.

Bacterial species exist that lack AHCY but instead express an ancient *SAH nucleosidase* that hydrolyses SAH into S-ribosylhomocysteine and adenine[7,8]. Surprisingly, partial rewiring of the mammalian methyl cycle by ectopically expressing the *E. coli SAH nucleosidase* completely protected the methyl cycle from AHCY inhibition, and allowed normal circadian rhythms, even in the presence of a saturating concentration of inhibitor. These observations demonstrate the importance of methyl metabolism in the regulation of biological rhythms from cyanobacteria to humans and suggest a therapeutic application of methyl cycle reprogramming to alleviate the detrimental impact of methylation deficiencies.

## Results

### AHCY is a highly conserved enzyme in the methyl cycle.
Methylation deficiency can be induced by carbocyclic adenosine analogs identified as AHCY inhibitors more than 30 years ago, such as 3-Deazaneplanocin A (DZnep)[9–11]. This pharmacological inhibition mimics the pathological symptoms caused by AHCY deficiency[12], such as high plasma methionine, SAM and SAH; all indicators of methyl cycle aberrations. AHCY catalyzes the cleavage of SAH to adenosine and L-homocysteine, and DZnep inhibits this reaction by occupying the adenosine binding site of AHCY. The crystal structures of human[13], mouse[14] and yellow lupin (*Lupinus luteus*)[15] AHCY complexed with adenosine or analogs have been described, and insights into its catalytic activity have been obtained[16–19]. AHCY is one of the most evolutionarily conserved proteins[20], but experimentally determined structures of AHCY complexed with DZnep remain to be described for most organisms. A full-length multiple sequence alignment (Supplementary Fig. 1) and homology modeling of AHCY from humans to cyanobacteria (Fig. 1a and Supplementary Movie 1) revealed high conservation of sequence and predicted tertiary structure.

Amino acids contributing to the DZnep binding site showed at least 88% identity between all eukaryotic AHCY sequences, and 78% between human and bacterial sequences, indicating that the DZnep binding site is virtually identical across a wide range of organisms (Fig. 1b, c and Supplementary Fig. 2). Moreover, amino acids that were reported as crucial for the activity of rat AHCY (His55, Asp130, Glu155, Lys186, Asp190, and Asn191[19]) are universal (Supplementary Fig. 1). In line with the above, molecular docking simulations of AHCY with adenosine or DZnep revealed comparable ligand binding conformations and estimated binding free energies for all organisms (Fig. 1d and Supplementary Fig. 2).

DZnep is sometimes erroneously considered to be a histone methyltransferase EZH2 inhibitor, because of the misinterpretation of a report showing that it inhibits histone methylation sites targeted by EZH2 in cancer cells[21]. More recent reports showed that DZnep widely inhibits histone methylation, in line with its inhibitory effect via AHCY[22,23]. Together these observations strongly support the use of DZnep as a valid approach to disrupt the methyl cycle across taxa, as we next demonstrate.

### Methylation deficiency disrupts the metazoan circadian clock.
To test the effects of DZnep on the mammalian clock, we exposed a human osteosarcoma cell line (U-2 OS) and PER2::LUC mouse embryonic fibroblasts (MEFs), expressing bioluminescent clock reporters[24,25], to DZnep concentrations. In human (Fig. 2a, b) and mouse (Fig. 2c, d) cells, DZnep potently and dose-dependently increased the period of circadian oscillations, visualized by bioluminescent rhythms of *Bmal1* or *Per2* expression, respectively.

We next tested DZnep on the zebrafish embryonic PAC2 cell line expressing a luminescent reporter of the *Per1b* gene, a non-mammalian cell type commonly used for circadian studies, and revealed, as observed in mammalian cells, a clear effect of the drug on the circadian period (Fig. 2e, f). An effect on the entrainment to the light/dark cycles (see methods) was also observed: The *Per1b* gene normally peaks at dawn but was severely blunted and delayed in DZnep-treated cells.

To test the effects of methylation deficiency on circadian rhythms in invertebrates, we exposed *Drosophila melanogaster* haltere cultures from transgenic luciferase reporter *ptim TIM-LUC* flies to DZnep. As observed in vertebrates, DZnep caused dose-dependent period lengthening (Fig. 2g, h). In addition, an effect of 100 μM DZnep on the luminescent rhythms reporting the expression of the *sur-5* gene was also observed in freely moving *C. elegans* (Supplementary Fig. 3), a relatively new model in circadian biology[26].

### Methylation deficiency also disrupts the somite segmentation clock.
While the metazoan circadian clock is an oscillator with a period near 24 h, the somite segmentation clock in mammalian embryos cycles much faster and orchestrates the appearance of new segments or "somites" from the paraxial mesoderm (Supplementary Fig. 4a). The underlying molecular oscillator is centered on the transcription factor *Hairy & Enhancer Of Split 7* (*Hes7*), the expression of which oscillates with a period close to 2 h in mouse[27]. Like circadian clock genes, *Hes7* oscillatory expression can be monitored in real-time from transgenic embryos expressing highly destabilized luciferase under the control of the *Hes7* promoter. Testing increasing concentrations of DZnep on these transgenic embryos revealed clear similarity to the effects of DZnep on circadian clock markers, resulting in the lengthening of the somite segmentation clock period (Supplementary Fig. 4b–d, Supplementary Movie 2). Together, these data demonstrate the importance of the methyl cycle in the regulation

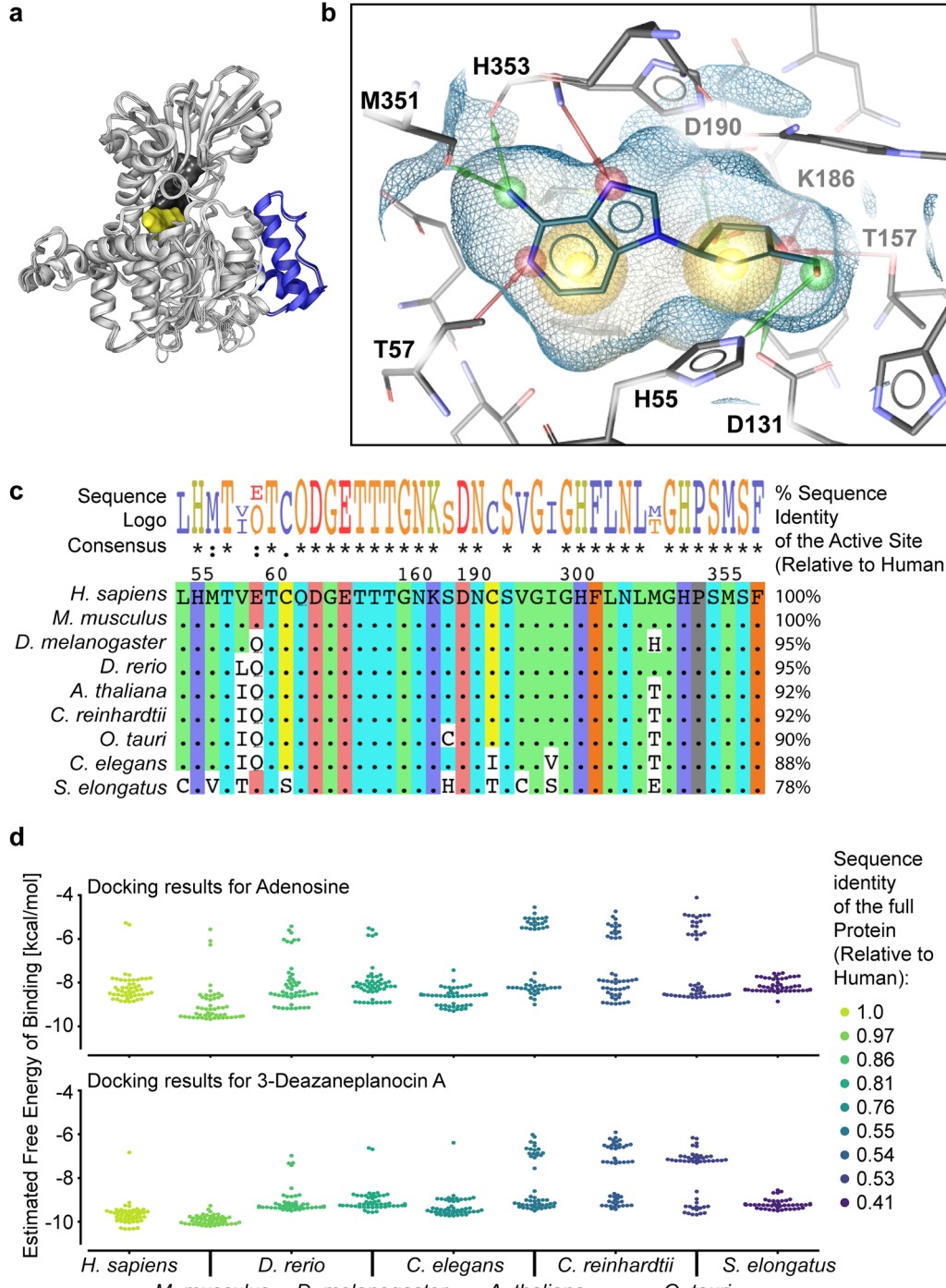

**Fig. 1 Adenosylhomocysteinase is a highly conserved protein. a** Structural superposition of AHCY from the 9 organisms investigated here, using human (1LI4), mouse (5AXA) or lupin (3OND) crystal structures as templates. The blue loop is specific to plants and green algae; DZnep is shown in yellow, NAD+ in black. See also Supplementary Movie 1. **b** Docking simulation of human AHCY with DZnep, based on the 1LI4 crystal structure of human AHCY complexed with Neplanocin A, an analog of DZnep. The amino acids involved in DZnep binding are indicated with their position. See Supplementary Fig. 2 for docking simulations of DZnep to AHCY from other organisms. Red and green arrows are hydrogen bonds, yellow spheres are hydrophobic effects. The estimated free energy of binding for depicted DZnep docking conformation was −9.87 kcal/mol. **c** Discontinuous alignment of amino acids contributing to the binding of DZnep, using the human sequence as a reference and with sequence identities shown on the right. When amino acids are identical to human, a dot is shown in the alignment. The sequence logo on top is a graphical representation of the conservation of amino acids, with the consensus symbols below (* = fully conserved residue, : = conservation of strongly similar properties, . = conservation of weakly similar properties). The positions of selected conserved amino acids are given for the human sequence on top of the alignment. **d** Molecular docking simulations of AHCY with adenosine and DZnep showing comparable binding free energies in all organisms. Colors represent full sequence identities, relative to human.

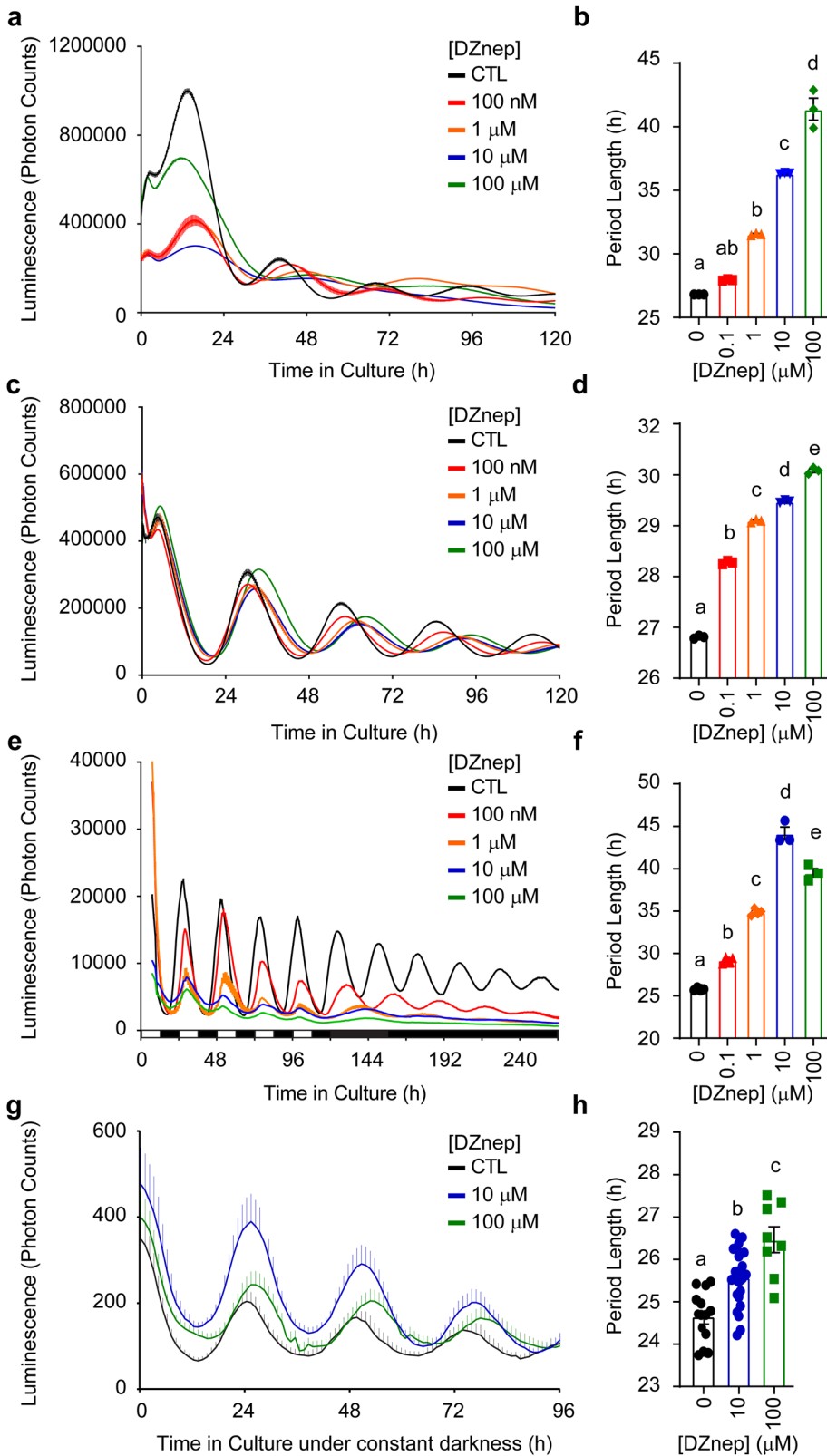

**Fig. 2 Circadian rhythms are a quantitative gauge for methylation deficiency in Metazoa. a** Mean luminescence ± SEM of human *Bmal1*-luc U-2 OS cells[25] treated with increasing concentrations of DZnep; **b** shows mean ± SEM of period, *n* = 3 dishes. **c** Mean luminescence ± SEM of PER2::LUC MEF[24] treated with DZnep, *n* = 3 dishes; mean ± SEM of period shown in **d**. **e** Mean luminescence ± SEM of *Per1b*-luciferase PAC-2 zebrafish cells treated with DZnep, *n* = 4 dishes; mean ± SEM of period shown in **f**; note the initial 96 h of data performed under light/dark cycles. **g** Mean luminescence ± SEM of *ptim*-TIM-LUC *Drosophila* halteres treated with DZnep, *n* = 8 independent halteres, with only the upper segment of the error bars shown for clarity; mean ± SEM of period shown in **h**. All bar graphs analyzed by One-Way ANOVA followed by Bonferroni's test; all indicated comparisons (**a** vs. **b** vs. **c** vs. **d** vs. **e**), at least *p* < 0.05.

of transcriptional programs orchestrating biological rhythms in Metazoa.

**Methylation deficiency disrupts the clock in plants and algae.** The most commonly used model organism to study circadian rhythms in land plants is *Arabidopsis thaliana*. We thus tested the effects of DZnep on luminescent rhythms reporting the expression of the plant evening gene TOC1 in protoplasts and extended our investigations to aquatic unicellular green algae. *Ostreococcus tauri* and *Chlamydomonas reinhardtii* represent two different classes of unicellular green algae that have successfully been used in circadian studies and constitute great models to investigate cell-autonomous metabolic processes[28–30]. We tested increasing concentration of DZnep on *Arabidopsis* (Fig. 3a, b), *Ostreococcus tauri* (Fig. 3c, d) and *Chlamydomonas reinhardtii* (Fig. 3f, g), and observed an increase in period length almost identical to the results obtained in vertebrates. A plant- and green algae-specific domain has been previously described in AHCY[31], from amino acids 151 to 191 (Fig. 1 and Supplementary Fig. 1). Despite this insertion, however, the domains for SAH and NAD$^+$ binding are remarkably conserved. We already showed that DZnep analogs increase SAH in mammalian cells[6], but to confirm that DZnep does increase SAH also in the green plant clade, we quantified SAM and SAH in DZnep-treated algae and revealed a dose-dependent increase in SAH and a decrease in methylation potential (Fig. 3e, h and Supplementary Fig. 5). In conclusion, the effects of DZnep treatment on transcriptional rhythms are also conserved in the plant kingdom. These results are especially meaningful given unicellular algae and humans are separated by more than 1 billion years of evolution[32].

**The methyl cycle in cyanobacteria is less sensitive to AHCY inhibition.** The circadian oscillator in cyanobacteria is composed of three proteins. Kai-A, -B and -C form a complex that regulates two key activities of KaiC: ATPase and autophosphorylation activity that result in an autonomous and self-sustained phosphorylation-based oscillator ticking with a period close to 24-h. In a cellular context, this non-transcriptional oscillator controls transcriptional outputs that in turn add robustness to the biochemical oscillator via transcription-translation feedback loops[33]. Despite these pronounced differences in clock architecture, the circadian period in cyanobacteria lengthened in response to low concentrations of DZnep (Fig. 4a, b), but with a much lower magnitude compared to that in the eukaryotes tested here.

Inhibition of methylations by DZnep depends on the unique activity of AHCY to directly hydrolyze SAH into Hcy and adenosine. In prokaryotes, the enzyme 5'-*methylthioadenosine/S-adenosylhomocysteine nucleosidase* (MTAN) has a SAH nucleosidase activity that cleaves SAH to adenine and S-ribosylhomocysteine[34]. This pathway for AHCY catabolism, acting as a buffer against SAH accumulation, could explain the weaker response to DZnep compared to eukaryotes. Indeed, MTANs have been identified in some *Synechococcus* strains, such as PCC7336 and MED-G69[35]. The lack of a strong response to DZnep may be due either to the presence of MTAN in *S. elongatus*, to DZnep being unable to bind to AHCY (this may be supported by the lowest sequence identity of AHCY in *S. elongatus* shown in Fig. 1) or having low bioavailability, or to the lack of methylation-dependent regulation of the cyanobacterial clock. Surprisingly, higher concentrations of the drug caused no significant effects on the period (Fig. 4b). Therefore, to independently verify the importance of methylations for cyanobacterial clock, we used the global methylation inhibitor sinefungin. This is a natural analog of SAM that directly binds to and inhibits methyltransferases[36]; sinefungin was previously used

in cyanobacteria to elicit methylation-dependent morphological changes[37]. Sinefungin was more potent than DZnep, causing a dose-dependent period lengthening of a much greater magnitude (Fig. 4c, d). These data reveal an unexpected role for methylation even in the cyanobacterial circadian clock, but the methyl cycle in these cells appeared more resilient to DZnep-induced deficiency, maybe because of an alternative pathway for SAH catabolism that eukaryotes lack (Fig. 4e).

**Bacterial SAH nucleosidase rescues mammalian cells from AHCY inhibition.** AHCY mutations (R49C, D86G, Y143C, W112X and A89V) in humans lead to greatly elevated levels of S-adenosylhomocysteine (>100-fold), causing early developmental stagnation and severe pathophysiology[12,38,39]. If the methyl cycle in these patients could be rewired to prevent S-adenosylhomocysteine build-up, it would protect them from AHCY deficiency. To test this potentially transformative hypothesis, we ectopically expressed the wild-type (WT) or the inactive D197A mutant MTAN from *E. coli*[8] in mouse and human cells. In itself, expression of the MTAN variants, which was confirmed in mouse and human cells by immunoblotting, did not significantly change circadian parameters (Supplementary Fig. 6a, b). Cells transfected with either vectors grew well and did not display any abnormalities, in line with S-ribosylhomocysteine being a meaningless metabolite in mammalian cells. Strikingly, a complete protection from the effects of DZnep was seen in human cells expressing WT MTAN, even at the saturating concentrations of 10 and 100 μM (Fig. 5a). D197A MTAN did not provide protection, indicating the specificity of the effect (Fig. 5b). Identical results were observed in mouse cells (Fig. 5c, d). Quantification of period in human and mouse cells clearly revealed this protective effect of WT MTAN (Fig. 5e, f). Mortality and/or cell growth inhibition was observed at high DZnep concentrations on cells actively growing, but WT MTAN protected the cells against this effect whereas D197A MTAN did not (Supplementary Fig. 6c–f). Quantification of SAM and SAH confirmed the DZnep-dependent increase in SAH and resultant collapse of the methylation potential in cells transfected with mutant, but not wild-type MTAN (Fig. 5g, Supplementary Fig. 6g, h). Interestingly, in vehicle-treated cells the expression of the wild-type MTAN was sufficient to increase the methylation potential compared to cells expressing the mutant enzyme (Supplementary Fig. 6g, h).

Finally, we sought to determine the effects of DZnep on global lysine and arginine methylation, on RNA and mRNA $N^6$-methyladenosine (m6A), and on DNA 5-methylcytosine (m5C), as well as the rescue of their potential inhibition by MTAN in mouse cells. In all cases, PER2::LUC MEFs were treated with DZnep 0, 5 or 10 μM (to obtain near-maximum period lengthening) for 48-hours.

Immunoblotting using an antibody against mono- and di-methylated lysine showed that a few proteins had lower methylated levels under DZnep treatment, but only in cells transfected with the mutant inactive MTAN (Fig. 5h). Probing mono- and di-methylated arginine in the same cells showed somewhat less pronounced methylation inhibition, which was also rescued by WT MTAN (Supplementary Fig. 6i). We also checked the global levels of the specific methyl marks Histone 4 Arginine 3 symmetric demethylation (transcriptional repression) and Histone H3 Lysine 4 trimethylation (transcriptional activation) but little inhibition was seen (Supplementary Fig. 6i).

$N^6$-methyladenosine did not significantly fluctuate between treatments when measured from total RNA samples, composed mainly (>90%) of 18S and 28S rRNA, each macromolecule

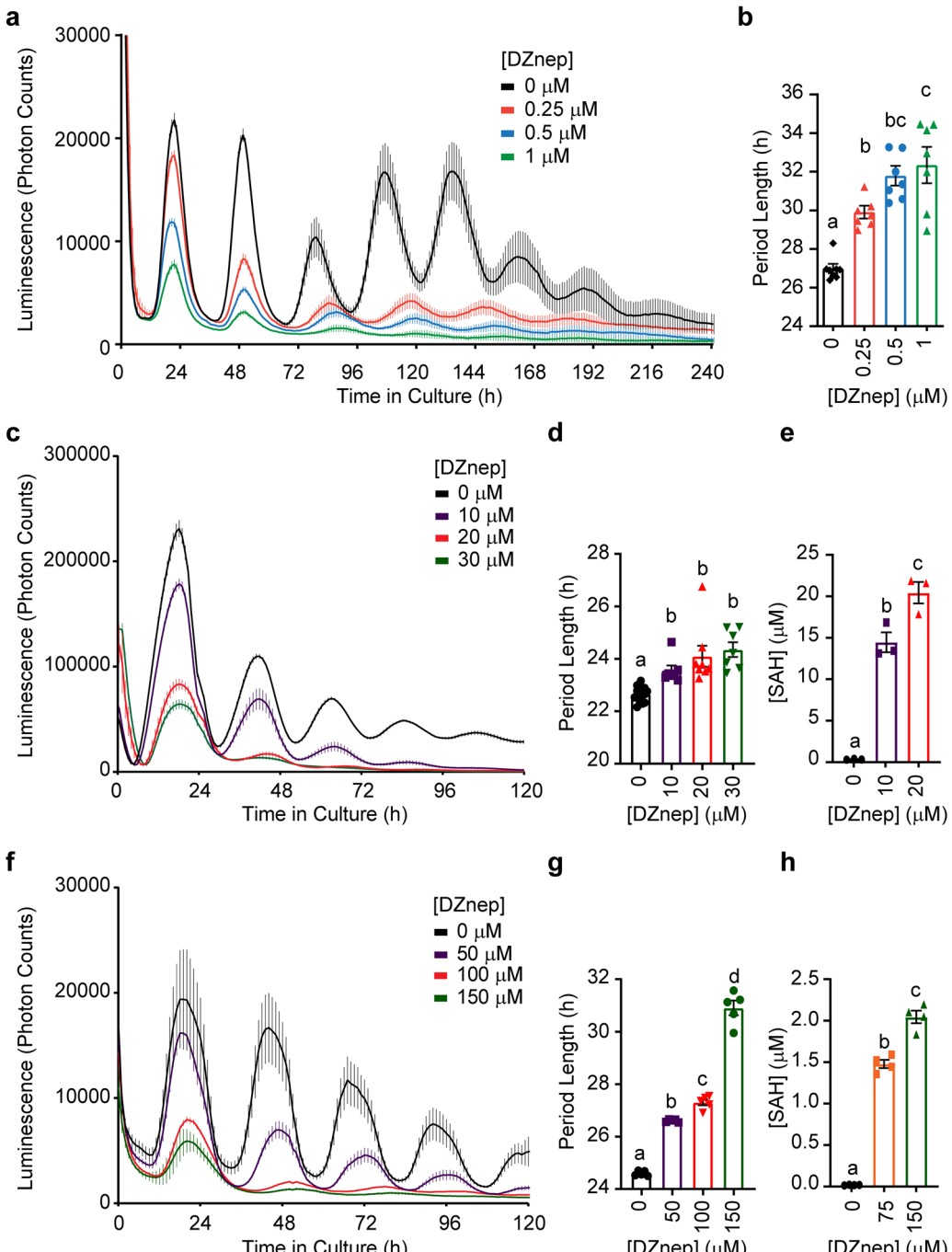

**Fig. 3 Biological rhythms are a quantitative gauge for methylation deficiency in plants and algae. a** Mean luminescence ± SEM of *Arabidopsis thaliana* CCA1pro:LUC protoplasts treated with different concentration of DZnep, $n = 8$ wells per treatment. For comparison between different runs, traces were aligned in relation to the first peak; **b** shows mean ± SEM of period, $n = 8$ wells. **c** Mean luminescence ± SEM of CCA1-LUC *Ostreococcus tauri* cells, $n = 7$ wells; **d** shows mean ± SEM of period, $n = 7$ wells. No significance was observed between 10, 20 and 30 μM, but the significance compared to 0 μM became stronger, i.e., $p < 0.05$, $p < 0.001$, $p < 0.0001$, respectively, indicating dose-dependent effects. **e** Mean SAH concentration ± SEM in *O. tauri* treated with the indicated concentrations of DZnep, $n = 3$ wells. **f** Mean luminescence ± SEM of *tuf*A-lucCP *Chlamydomonas reinhardtii* CBR cells treated with DZnep, $n = 5$ wells per treatment; **g** shows mean ± SEM of period, $n = 5$ wells. **h** Mean SAH concentration ± SEM in *C. reinhardtii* treated with the indicated concentrations of DZnep, $n = 4$ wells. All bar graphs analyzed by One-Way ANOVA (all $p < 0.0001$) followed by Bonferroni's test; all indicated comparisons (**a** vs. **b** vs. **c** vs. **d**) at least $p < 0.05$. See also Supplementary Fig. 5.

containing only one single m6A site (Supplementary Fig. 6j). When quantified from mRNA, however, a decrease in m6A was detected under DZnep treatment, but only in cells transfected by the mutant inactive MTAN (Fig. 5i, j, k). In contrast, DNA m5C was not significantly affected (Supplementary Fig. 6k).

It is possible that a longer treatment with DZnep may have affected more stable methylations such as m6A in rRNA and m5C in DNA or caused a more widespread inhibition of histone methylation, but since the period lengthening effects of DZnep are observable by 24 h we speculate these methylations may not contribute to the period lengthening. Promoter-specific DNA or

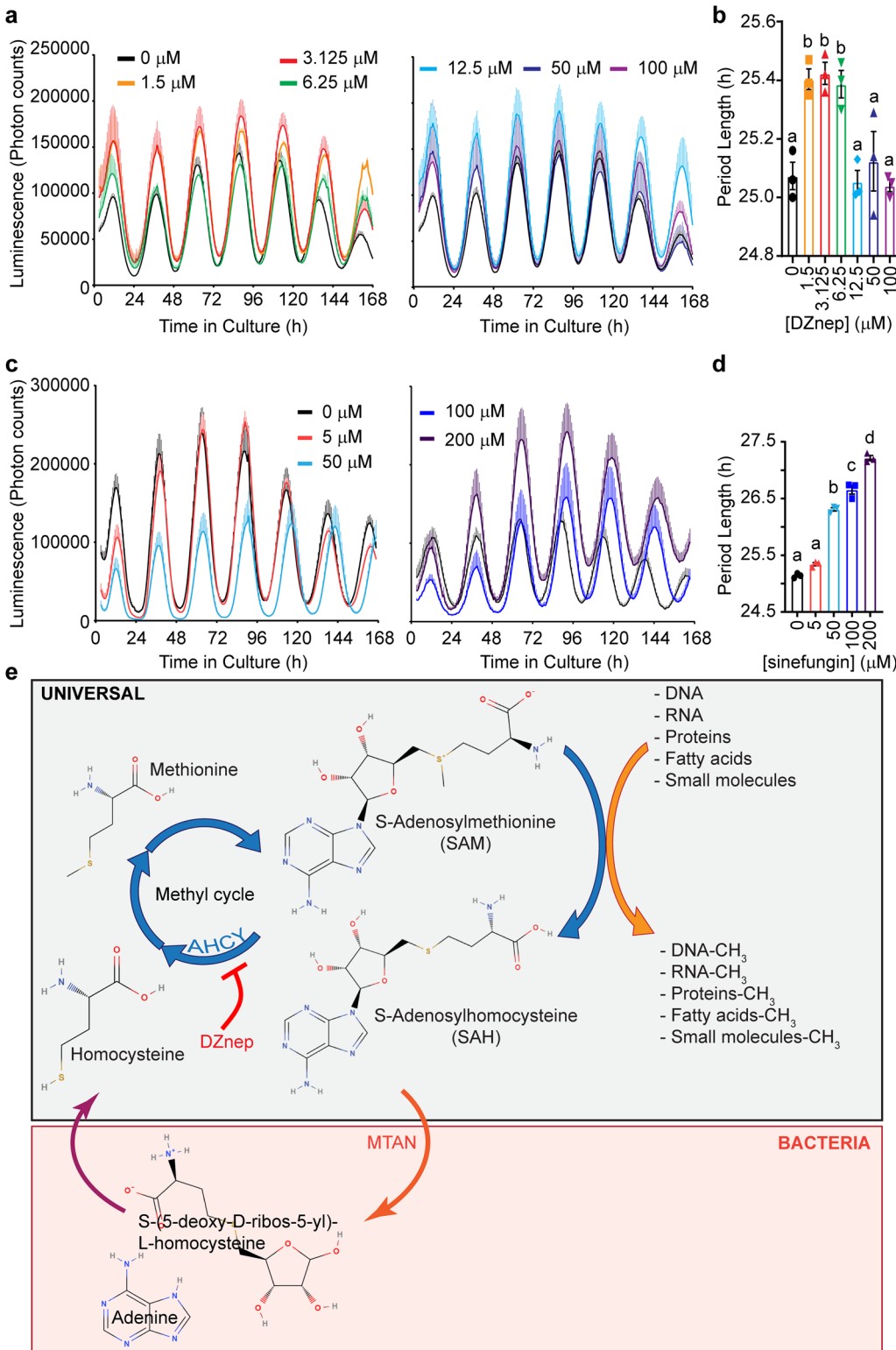

**Fig. 4 Cyanobacteria are less sensitive to AHCY inhibition. a** Mean luminescence ± SEM of *Synechococcus* PCC 7942 *kaiB*Cp::luxAB knock-in strain treated with different concentrations of DZnep, *n* = 3 colonies, with only the upper section of the error bars and higher DZnep concentrations shown on a different graph for clarity; **b** shows mean period ± SEM, *n* = 3 colonies. **c** Mean luminescence ± SEM of *Synechococcus* PCC 7942 *kaiB*Cp::luxAB treated with different Sinefungin concentrations as indicated over the graphs; **d** shows mean period ± SEM, *n* = 3 colonies. **e** Organization of the methyl cycle, with the two-steps SAH conversion to homocysteine in bacteria, starting with MTAN. One arrow represents one enzyme, except for methyltransferases that all use SAM as a co-substrate to methylate different targets, generating SAH in the process. The enzyme AHCY, mediating SAH hydrolysis and inhibited by DZnep, is indicated on the picture. All bar graphs analyzed by One-Way ANOVA followed by Bonferroni's test; all indicated comparisons (**a** vs. **b** vs. **c** vs. **d**) at least $p < 0.05$.

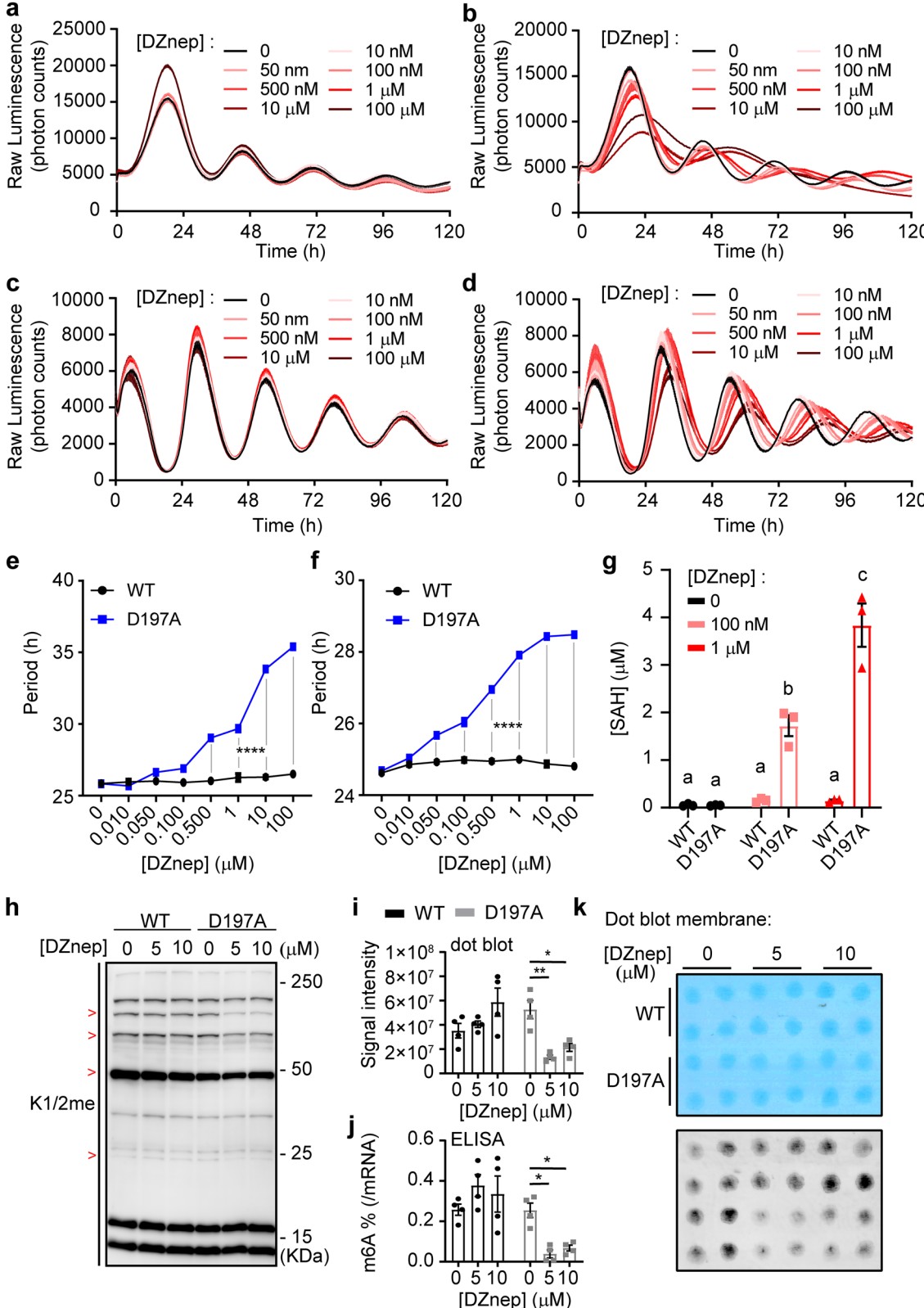

histone methylation, for example in the promoter of clock genes, which would not be detectable when global methylation is quantified, may also be inhibited by DZnep.

In conclusion, here we have revealed the evolutionarily conserved link between methyl metabolism and biological clocks. Moreover, we have demonstrated in mammalian cells that the period lengthening effects of DZnep exclusively depends on the inhibition of AHCY and the increase in SAH, mainly leading to protein and mRNA m6A methylation inhibition. We propose that the partial rewiring of the mammalian methyl cycle, protecting proteins and nucleic acids from the consequences of methyl cycle inhibition, might present a previously unreported strategy to treat methylation deficiencies, such as homocysteinemia or the autosomal recessive AHCY deficiency[12,38–41].

**Fig. 5 Rewiring the methyl cycle protects mammalian cells from methylation deficiency. a, b** Mean luminescence ± SEM of human *Bmal1*-luc U-2 OS cells transfected with WT and mutant MTAN, respectively, and treated with different concentrations of DZnep, $n = 3$ dishes per treatment. **c, d** Mean luminescence ± SEM of PER2::LUC MEFs transfected with WT and mutant MTAN, respectively, and treated with different concentrations of DZnep, $n = 3$ dishes per treatment. **e, f** Mean period ± SEM of *Bmal1*-luc U-2 OS cells and PER2::LUC MEFs from **a, b** and **c, d**, respectively, analyzed by two-way ANOVA followed by Bonferroni's test, $n = 3$ dishes per treatment. The gray bars indicate which comparisons reach significance, all $p < 0.0001$ (****). **g** Mean SAH concentration ± SEM in DZnep-treated PER2::LUC MEFs, $n = 3$ dishes per treatment, analyzed by Two-Way ANOVA (all sources of variations $p < 0.0001$) followed by Bonferroni's test; all indicated comparisons (**a** vs. **b** vs. **c**) at least $p < 0.05$. See also Supplementary Fig. 6. **h** Immunoblots using an antibody against mono- and di- methylated lysine (K1/2me) reveal that the methylation of some proteins is inhibited by DZnep, and rescued by WT but not D197A MTAN. The red > signs indicate such proteins. Actin loading control is shown in Supplementary Fig. 6i. **i, j** Mean mRNA m6A methylation levels ± SEM in PER2::LUC MEFs transfected with WT or D197A MTAN, then treated with DZnep for 48 h; **i, j** showing dot blot assay and enzyme-linked immunosorbent assay-based quantification (ELISA), respectively. Dot blot and ELISA analyzed separately by Two-Way ANOVA (DZnep treatment, MTAN effect and interactions at least $p < 0.05$) followed by Bonferroni's test (*$p < 0.05$; **$p < 0.01$), $n = 4$ dishes. **k** Dot blot membrane quantified in **i**, with 150 ng mRNA/dot, first stained with methylene blue (top) as a loading control.

## Discussion

Solar ultraviolet radiation played a critical role in prebiotic chemistry, but the energy of UV photons can also degrade biologically important molecules, preventing abiogenesis. Once autocatalytic pseudo-life forms evolved with more complex chemistry, avoidance mechanisms may have provided increased fitness in such an environment. The origin of the circadian clock on the early Earth is likely to have been a simple sensing of molecules present in the milieu, increasing or decreasing under UV exposure and triggering an appropriate response. Methyl cycle metabolites are thought to have been present in the prebiotic world, since methionine can be created by a spark discharge and is proposed to be an intermediate in the prebiotic synthesis of Hcy, which was also found among organic molecules synthesized in the 1972 Miller experiment[42]. In addition, formaldehyde, an abundant prebiotic molecule increasing during the day under the action of UV on carbon monoxide and water vapor[43] could have affected the chemistry of early life-forms by methylation[44] and may have worked like a chemical marker of daylight. A conserved link between methyl transfer and the circadian clock may have arisen from such a scenario.

In mammalian cells we previously reported that methyl cycle inhibition decreases the methylation of internal adenosines in mRNA (m6A), as well as that of histones[6]. While specific mRNA m6A inhibition was sufficient to elicit period elongation, histone methylation likely contributed to the period elongation obtained by DZnep, as well as that of other methylation sites in mRNA, rRNA and tRNA. Due to the considerable heterogeneity in mechanisms regulating gene expression and function in organisms tested here, identifying a single mechanism explaining period elongation in all taxa would be a difficult undertaking and is beyond the scope of the present work. Since the oldest methyltransferases are RNA methyltransferases[45], it is tempting to propose that inhibition of RNA methylation may at least partially contribute to the period lengthening in all species.

That mammalian cells expressing *E. coli* MTAN are completely protected from the period-lengthening effects of DZnep demonstrates that these effects are solely dependent upon methyl cycle inhibition and SAH accumulation. Moreover, we have shown in mouse cells that the functional consequences of methyl cycle inhibition, *i.e.*, a decrease in the methylation of proteins and mRNA, is also rescued by WT MTAN. This confirms that MTAN, despite removing SAH from the cycle in a way that does not normally occur in mammalian cells, allows the methyl cycle to turn and drive cellular methylations when AHCY does not function. MTAN hydrolyses SAH to S-ribosylhomocysteine and adenine, not to Hcy and adenosine as AHCY does. Since Hcy is needed for the regeneration of methionine in the methyl cycle, how could the methyl cycle still maintain cellular methylations active without Hcy? It is likely that, with the abundance of free

methionine in the cell, the synthesis of SAM and downstream methylations can continue unimpaired even when Hcy is low. Hcy is however also used for the synthesis of glutathione (GSH) via cystathionine and cysteine, a metabolic route called the transsulfuration pathway (Hcy → cystathionine → cysteine → GSH), which could potentially have been inhibited by MTAN expression and DZnep. To investigate potential disruption of this pathway caused by either DZnep or MTAN, we quantified these metabolites from cell extracts used in Fig. 5g. As expected, expression of WT MTAN caused a decrease in Hcy compared to cells expressing D197A (Supplementary Fig. 7). Overall, however, the transsulfuration pathway appeared more active in cells expressing WT MTAN, especially given the higher concentrations of cysteine and GSH, which were probably due to the increase in SAM observed under the same conditions (Supplementary Fig. 6g), SAM being a powerful allosteric activator of the transsulfuration pathway[46]. A general tendency of DZnep to inhibit the transsulfuration pathway was observed, more pronounced in cells expressing the mutant MTAN (Supplementary Fig. 7). Glutathione disulfide (GSSG), generated during enzyme-catalyzed reduction reactions of GSH with peroxides or disulfide bonds, tended to decrease under DZnep treatment (Supplementary Fig. 7). The GSH/GSSH ratio, a measure of oxidative stress, was not significantly affected by either DZnep or MTAN, indicating that the mild effects of DZnep and MTAN expression on the transsulfuration pathway are unlikely to contribute to the circadian period lengthening (Supplementary Fig. 7). These data do show that WT MTAN also rescues the cells from the mild inhibition of the transsulfuration pathway by DZnep, however.

The somite segmentation clock was strongly affected by inhibition of AHCY, which is in line with the importance of 1 carbon metabolism for embryonic development. As can be seen in the Supplementary Movie 3, showing similar results to Supplementary Movie 2 but merged with brightfield images and displaying luminescence as a pseudo-color green, period lengthening of the oscillatory expression of *Hes7-luciferase* occurred together with a pronounced delay in the growth of the caudal tip of the presomitic mesoderm as well as in the appearance of new somites, demonstrating that this molecular clock as well as its output, *i.e.* the somitogenesis, were affected by DZnep. The mechanisms underlying this period lengthening may be distinct from those involved in the lengthening of the circadian period. Considering the short period of *Hes7* oscillations—2 h—, and the importance of 3'-UTR-dependent regulation of *Hes7* mRNA turnover for its cyclic expression[47], the inhibition of mRNA methylation may at least in part contribute to the results observed.

We have presented here an identical response to methyl cycle inhibition on biological oscillators across a wide variety of species. Only a small number of examples have been reported that identified aspects of circadian timekeeping shared among

rhythmic life, such as oxidation cycles of peroxiredoxin[29,48,49] and regulation by oscillating intracellular magnesium concentrations[50]. Building on those early observations, this study establishes methylation as a universal regulator of rhythmicity. Importantly, to our knowledge we provide a first potential avenue of translation of fundamental understanding of methylation deficiencies to clinical applications. Biopharmaceutics, such as recombinant MTAN or therapies using genetically engineered cells expressing MTAN, could be used to treat methylation deficiencies in the short and longer term. We consider this superior to approaches directly targeting deficient AHCY in cases where the deficiency arises from errors of metabolism causing accumulation of metabolites that are inhibitors of AHCY[12]; and even in cases of AHCY polymorphisms, since the endogenous inactive AHCY in patients would compete with the exogenous active AHCY.

## Methods

**Molecular modeling and sequence analyses**. 3D protein structures were obtained from the Protein Data Bank (PDB) for *Homo sapiens* (PDB 1LI4[13], S-adenosylhomocysteine hydrolase complexed with neplanocin, resolution 2.01 Å) and *Mus musculus* (PDB 5AXA[14], S-adenosylhomocysteine hydrolase with adenosine, chain A, resolution 1.55 Å). All PDB structures underwent preprocessing including the addition of missing amino acid atoms and hydrogens, removal of solvent atoms, and hydrogen-bond network optimization and residue protonation based on predicted pKa values using the Protein Preparation Wizard[51] of Maestro Schrödinger Release 2018-1 [Maestro, Schrödinger, LLC, New York, NY, 2018]. Homology models of *Danio rerio* (template 1LI4, 100% coverage, 86% sequence identity), *Drosophila melanogaster* (template 1LI4, 100% coverage, 81% sequence identity), *Caenorhabditis elegans* (template 1LI4, 98% coverage, 77% sequence identity), *Arabidopsis thaliana* (template 3OND[15] chain A, resolution 1.17 Å, 100% coverage, sequence 92% identity), *Chlamydomonas reinhardtii* (template 3OND chain A, 100% coverage, sequence 80% identity), *Ostreococcus tauri* (template 3OND chain A, 99% coverage, 79% sequence identity), and *Synechococcus elongatus* (template 1LI4, 98% coverage, 41% sequence identity) were constructed via the SWISS-MODEL server[52]. Model quality was evaluated based on GMQE (global model quality estimation) ranging from zero to one with high numbers indicating high model reliability, and QMEAN[53], a composite and absolute measure for the quality of protein models, indicating good or poor agreement between model and template with values of around 0 or below −4, respectively. GMQE and QMEAN values for the seven homology models were determined to be 0.98 and 0.85 for *D. rerio*, 0.93 and 0.82 for *D. melanogaster*, 0.87 and 0.35 for *C. elegans*, 0.99 and 0.31 for *A. thaliana*, 0.95 and 0.08 for *C. reinhardtii*, 0.91 and −0.54 for *O. tauri*, and 0.75 and −0.59 for *S. elongatus*, respectively. GROMACS version 2018[54] was utilized for energy minimization of all nine protein structures including oxidized cofactor $NAD^+$ using force field AMBERff99SB-ILDN[55]. Cofactor parameters and topologies were obtained through ACPYPE[56] using the AM1-BCC method with net charge $q = −1$. The systems including TIP3P water were neutralized with counter ions ($Cl^-$ and $Na^+$) to 0.1 M and minimized via steepest descent with a maximum force limit of 100 kJ/mol/nm. Molecular docking simulations were performed using the energy-minimized protein-$NAD^+$ complexes as well as co-crystallized adenosine (ADN), neplanocin (NOC), and 3-deazaneplanocin A (DZnep, constructed from co-crystal NOC in 1LI4[13]) using the Lamarckian Genetic Algorithm provided by the AutoDock4.2 suite[57]. Protein-ligand interactions were analyzed in LigandScout version 4.2[58,59].

Multiple sequence alignment of AHCY was performed by CLUSTALW[60] with the multiple sequence viewer from the Maestro module using the following reference sequences: for human, Genbank: NP_000678.1; mouse, Genbank: NP_057870.3; zebrafish, Genbank: NP_954688.1; fruit fly, Genbank: NP_511164.2; *C. elegans*, Genbank: NP_491955.1; *Arabidopsis*: NP_193130.1; *Chlamydomonas*, Genbank: XP_001693339.1; *Ostreococcus*, Genbank: XP_022839640.1; cyanobacteria, Genbank: WP_011243218.1.

For the molecular structure of drugs shown in Fig. 4e, the Open Source web application MolView was used.

**Assay for effect of the inhibition of the methyl cycle on vertebrate circadian clock**. Human U-2 OS cells stably transfected with a *Bmal1*-luciferase reporter vector[25] and mouse PER2::LUC MEFs[24] cell lines were cultivated as previously described[6]. Briefly, cells were seeded onto 35 mm dishes (Corning, New York, USA) and allowed to grow to confluence in DMEM/F12 medium (Nacalai, Kyoto, Japan) containing penicillin/streptomycine/amphotericin (Nacalai). Cells were then shocked by dexamethasone (Sigma-Aldrich, St. Louis, MO, USA) 200 nM for 2 h, followed by a medium change including 1 mM luciferine (Nacalai). 35 mm dishes were then transferred to an 8-dishes luminometer-incubator (Kronos Dio, Atto, Tokyo, Japan). Photons were counted in bins of 2 min at a frequency of 10 min.

DZnep (SML0305) was purchased from Sigma-Aldrich. Period was estimated by BioDare2[61].

For zebrafish cell line and bioluminescence assays, the generation of a per1b-luciferase cell line from zebrafish PAC2 cells has been previously described[62]. Cells were cultured in Leibovitz's L-15 medium (Gibco, Thermo Fisher Scientific, Waltham, MA, USA) containing 15% fetal bovine serum (Biochrom AG, Berlin, Germany), 50 U/mL penicillin/streptomycin (Gibco), and 50 µg/mL gentamicin (Gibco). Cells were seeded at a density of 50,000-100,000 cells per well in quadruplicate wells of a 96-well plate in medium supplemented with 0.5 mM beetle luciferin (Promega, Madison, Wisconsin, USA), and drugs were prepared in water and added at the concentrations indicated in the figures and legends. Plates were sealed with clear adhesive TopSeal (Perkin Elmer, Waltham, MA, USA). Since the circadian clock in these cells is directly light-sensitive and can be entrained by light-dark cycles, cells were synchronized to a 12:12 LD cycle for 7 days and then transferred into DD for at least 3 days. Bioluminescence was monitored at 28°C on a Packard TopCount NXT scintillation counter (Perkin Elmer). DZnep (SML0305) was purchased from Sigma-Aldrich. Period was estimated by BioDare2[61].

**Assays for the inhibition of DNA, total RNA, mRNA and protein methylation**. Global DNA methylation was quantified by the Abcam 5-methylcytosine assay kit (ab233486, colorimetric) following manufacturer's protocol.

Global total RNA and mRNA m6A was quantified by dot blot assay as described below, and mRNA m6A was further confirmed by the Abcam m6A RNA Methylation Assay Kit (ab185912, colorimetric) following manufacturer's protocol. For the dot blot assay, positively charged nylon membrane was first bathed in DEPC-treated water for 10 min followed by DEPC-treated 6× SSC buffer for at least 15 min. During that time, RNA samples were denatured at 95 °C for 3 min and directly transferred on ice. Membranes were then laid on top of two flat layers of blotting paper wet but not dripping with 6× DEPC-treated SSC buffer, the remaining SSC on top of the membrane allowed to seep through it. 3 µl of each RNA sample (50 ng/µl for mRNA or 400 ng/µl for totRNA) were then spotted on the membrane and allowed to seep through. RNA was crosslinked to the membrane in a FUNA-UV crosslinker (Funakoshi, Japan) equipped with 254 nm UV lights, set to "optimal crosslink" (120,000 microjoules/cm²). Membrane was rinsed with DEPC-treated water then transferred to 0.02% Methylene Blue in 0.3 M Na acetate for 15 min in order to confirm the presence and consistent RNA amount for each dot. After several washes with DEPC-water, membrane was scanned for record. The membrane was then washed with 0.1%-tTBS (Nacalai Tesque, Japan) for 10 min, then blocked for 1 h in 5% skimmed milk in 0.1%-tTBS. The membrane was incubated overnight at 4 °C with the anti-m6A antibody (202 003, Synaptic System, Germany) diluted 1:1000. The next day the membrane was washed three times 10 min with tTBS then incubated 1 h at RT with the secondary antibody (NA934, GE Life Science, USA) diluted 1:5000. After three washes of 10 min with tTBS, the membrane was incubated with ECL Prime reagent (GE Life Science) and imaged with a LAS-4000 imaging system (Fujifilm, Japan).

Protein methylation was measured by standard immunoblotting using the following antibodies and dilutions: Abcam ab23366, 1:500, for mono- and di-methylated lysine, Abcam ab5823, 1:500, for histone 4 Arginine 3 symmetric demethylation, Abcam ab8580, 1:1000, for Histone H3 Lysine 4 trimethylation, Sigma A5441, 1:10000, for actin, and Abcam ab412 clone 7E6, 1:500, for monomethylated/dimethylated arginine. The second antibody was GE Life Science NA934, 1:5000.

**Transfection of mammalian cells with expression vectors before luminometry**. The bacterial expression vector pPROEX HTa containing wild-type MTAN[8] was used as a template to amplify a full length MTAN with flanking SalI/NheI restriction sites at the 5′ and 3′ end, respectively. A single 5′ HA tag 5′-gga-tacccatacgacgtcccagactacgct-3′ was also facultatively inserted during polymerase chain reaction (PCR). The SalI/NheI digested, tagged or untagged PCR products were then ligated into a SalI/NheI-digested pSELECT-HYGRO-MCS (Invivogen, San Diego, California, USA) using Ligation High V2 (Toyobo, Osaka, japan). DH5apha *E. coli* (Takara Bio, Kusatsu, Japan) were transformed with the ligation products and streaked on a selective plate containing 100 microg/ml hygromycin. Colonies were screened by Sanger sequencing. The D197A mutation was then performed, with the wild-type vector as a template, by PCR using the primers 5′-CCATCTCCGCCGTGGCCGATC3′ and 5′-GATCGGCCACGGCGGAGATGG-3′ containing the mutated GCC codon, together with the 5' and 3' primers used for the cloning of MTAN. D197A MTAN was cloned into pSELECT-HYGRO-MCS as described above for the wild-type version.

Wild-type or D197A tagged MTAN vectors were transfected into PER2::LUC MEFs or U-2 OS in 24-well plates at ~90% confluence with Lipofectamine LTX and PLUS reagent (Invitrogen, Thermo Fisher Scientific) following manufacturer's protocol, incubating cells 12–24 (MEFs) or 6–8 (U-2 OS) hours with the transfection mixes. After transfection, medium was changed, and cells were incubated for 24–48 hours. Antibiotic/antimycotic was omitted during transfection and in the first medium change. Expression of the HA-tagged WT and D197A MTAN was confirmed by immunoblotting using an anti-HA antibody (Roche 11867423001, 1:1000) (Supplementary Fig. 6a and b). For the experiments shown in Supplementary Fig. 5c–f, transfected confluent cells were trypsinized and plated and very low density and allowed to grow a few days until ~30% confluence. Cells

were treated with dexamethasone 200 nM for 2 h for synchronization and, after a medium change (with 1 mM luciferin and various concentrations of DZnep), transferred to a luminometer/incubator (CL24A-LIC, Churitsu Electric Corp., Nagoya, Japan) set at 35 °C, 5% $CO_2$. Photons were counted in bins of 10 s at a frequency of 10 min.

**Preparation of PER2::LUC cells for SAM and SAH quantification.** Cells plated in 10 cm culture dishes at 90% confluence were transfected with wild-type or D197A MTAN vectors using lipofectamine LTX with PLUS reagent following manufacturer's protocol (Invitrogen). After 12–24 h of incubation with the transfection mixes, medium was changed once, omitting antibiotic/antimycotic, and cells were incubated for one day. Cells were then treated with different concentrations of DZnep and incubated for 48 h. Cells were washed once with 10 ml, then once with 2 ml 5% mannitol (Nacalai). Mannitol was thoroughly removed, and 0.8 ml methanol was added to the dish, rocking vigorously for 30 s for the methanol to homogenously cover the cells. Dishes were tipped and 0.55 ml milliQ containing 125 ng/ml BIS-TRIS (Sigma-Aldrich) was added directly to the methanol. Dishes were once more rocked by hand for 30 s, then the methanol/MilliQ mix (~0.9 ml due to methanol evaporation) was transferred to a 1.5 ml tube. Two 0.4 ml aliquots of the extracts were filtered through VIVASPIN 500 3KDa cut-off filters (Sartorius AG, Göttingen, Germany) then kept at −80 °C until quantification. Dishes containing cells plated simultaneously and submitted to the same treatment were used to calculate cell counts and total cell volume for normalization.

**Preparation of *O. tauri* and *C. reinhardtii* for SAM and SAH quantification.** *O. tauri* or *C. reinhardtii* were cultivated in their respective culture medium as described in the corresponding sections on methyl cycle inhibition in these cells. Cells were then treated with DZnep at the indicated concentration at late log phase. After 48 h, cells were harvested (~$3 \times 10E^7$ *C. reinhardtii* and ~$10E^9$ *O. tauri*) by centrifugation. Cell pellets were then resuspended with 5% mannitol (Nacalai) for *C. reinhardtii* or 1M sorbitol (Sigma-Aldrich) for *O. tauri*, which was then thoroughly removed after centrifugation; this wash was performed twice. Cell pellets were frozen in liquid nitrogen and kept at −80°C before analysis. Cell pellets were resuspended in 0.4 ml methanol (Nacalai), then 0.275 ml of MilliQ containing 125 ng/ml BIS-TRIS (Sigma-Aldrich) was added to the methanol. Cell resuspensions were submitted to 10 freeze-thaw cycles (liquid nitrogen/4 °C) to break the cell wall, with full-speed vortexing for 5 s between cycles. After the last thaw, samples were vortexed at full speed for 10 seconds then centrifuged at 12,000 $g$ at 4 °C for 10 min. Two 0.3 ml aliquots of supernatant were transferred to VIVASPIN 500 3KDa cut-off filters (Sartorius) and centrifuged as recommended by the manufacturer. Eluates were kept at -80°C until quantification by LC/MS/MS. Total cell counts ($3.01–4.34 \times 10^7$) or optical density (0.514–0.646) were used as normalizers for *Chlamydomonas* or *Ostreococcus*, respectively.

**Hcy, cystathionine, cysteine, GSH, GSSG, SAM and SAH quantification by LC/ MS/MS.** The standard stocks of SAM (Cayman Chemical, Ann Arbor, MI, USA) and SAH (Sigma-Aldrich) were dissolved in water (1 mg/mL) and stored at −30 °C until use. For the construction of calibration curves, a series of dilutions of SAM and SAH mixtures were prepared at the concentrations of 0.001, 0.01, 0.05, 0.1, 0.5, 1, 10, 50, 100, and 200 μM.

To quantify the concentrations of SAM and SAH using the multiple reaction monitoring (MRM), LC/MS/MS conditions were optimized. The LC/MS/MS quantifications were conducted using a LC-ESI-TQMS (LCMS8030plus, Shimadzu, Kyoto, Japan). The target product ions were determined as $m/z$ 250.10 for SAM (CE −15 V), $m/z$ 136.05 for SAH (CE −20 V) from the precursor ions of $[M + H]^+$ ($m/z$ 399 for SAM, $m/z$ 385 for SAH).

The LC conditions were designed as previously described[6], with some modifications: The elution program was set as a linear gradient of 95–10% of MeCN in water containing 0.1% HCOOH (0–15 min) followed by isocratic 10% MeCN in water containing 0.1% HCOOH (7 min), at a flow rate of 0.2 mL/min. ZIC-HILIC (2.1 × 150 mm, Merck Millipore, Burlington, MA, USA) was selected as an analytical column at a temperature of 40 °C.

For the quantification, a dilution series of SAM and SAH mixtures (3 μL injection) were subjected to MRM quantification as described above. The plotted peak areas were calculated by LabSolutions 5.9.7 (Shimadzu). The calibration curves prepared were $y = 367,000 (\pm 6019) \times$ for SAM (P-value <0.0001), $y = 105,100 (\pm 1395) \times$ for SAH (P-value <0.0001). The SAM and SAH concentrations of analytes (3 μL injection) were quantified using the calibration curves. All standards and samples were analyzed in triplicates.

For the quantification of Hcy, cystathionine, cysteine, GSH, and GSSG, the standards of L-Hcy (Sigma-Aldrich), L-cystathionine (Fujifilm-Wako, Osaka, Japan), L-cysteine (Tokyo Chemical Industry, Tokyo, Japan), GSH (Fujifilm-Wako), and GSSG (Fujifilm-Wako) were purchased. For the construction of calibration curves, a series of dilutions of Hcy, cystathionine, cysteine, GSH, and GSSG mixtures were prepared just before use at the concentrations of 0.01, 0.1, 1, 10, 50, and 100 μM.

The MRM quantification were conducted by the same platform described above. The target product ions were determined as $m/z$ 89.90 for Hcy (CE −10 V), $m/z$ 134.00 for cystathionine (CE −15 V), $m/z$ 59.00 for cysteine (CE −23 V), $m/z$

178.90 for GSH (CE −14 V), and $m/z$ 355.10 for GSSG (CE −24 V) from the precursor ions of $[M + H]^+$ ($m/z$ 136 for Hcy, $m/z$ 223 for cystathionine, $m/z$ 122 for cysteine, $m/z$ 308 for GSH, and $m/z$ 613 for GSSG). Since a divalent ion of $[M + 2 H]^{2+}$ of GSSG ($m/z$ 307) was also abundant as well as an ion of $[M + H]^+$ GSSG ($m/z$ 613) at full ion scan mode, the divalent ion was also chosen as precursor ion to obtain target product ion of $m/z$ 84.10 (CE −26 V).

The calibration curves prepared were $y = 17,580 (\pm 153.7) \times$ for Hcy (P-value <0.0001), $y = 6,444 (\pm 50.6) \times$ for cystathionine (P-value <0.0001), $y = 2,673 (\pm 33.0) \times$ for cysteine (P-value <0.0001), $y = 72,710 (\pm 538.8) \times$ for GSH (P-value <0.0001), $y = 8,206 (\pm 133.0) \times$ for $[M + H]^+$ of GSSG (P-value <0.0001), and $y = 14,100 (\pm 211.0) \times$ for $[M + 2 H]^{2+}$ of GSSG (P-value <0.0001). The concentrations of analytes (3 μL injection) were quantified using the calibration curves. All standards and samples were analyzed in triplicates.

**Assay for effect of the inhibition of the methyl cycle on invertebrate circadian clock.** Halteres of two- to seven-day old transgenic *ptim*-TIM-LUC males[63] kept under 12 h : 12 h light:dark cycles (LD) at 25°C were bilaterally dry dissected. Each pair was transferred into one well of a 96 well plate (Topcount, Perkin Elmer) filled with medium containing 80% Schneider's medium (Sigma-Aldrich), 20% inactivated Fetal Bovine Serum (Capricorn Scientific, Ebsdorfergrund, Germany) and 1% PenStrep (Sigma-Aldrich). Medium was fortified with 226 μM Luciferin (Biosynth AG, Staad Switzerland) and supplemented with 10, 100, or 250 μM Dznep A (Sigma-Aldrich SML0305) diluted in PBS. Plates were sealed with clear adhesive covers and transferred to a TopCount plate reader (PerkinElmer). Bioluminescence emanating from each well was measured hourly in LD for two days, followed by 5 days of constant darkness (DD) at 25 °C as previously described[64]. The *ptim*-TIM-LUC reporter contains the timeless (*tim*) promoter sequences driving rhythmic expression of the *tim* cDNA, which is fused to the firefly luciferase cDNA. Period was estimated by BioDare2[61].

*C. elegans* strain N2 (Bristol strain, wild-type was provided by the *Caenorhabditis* Genetics Center, University of Minnesota (cbs.umn.edu/cgc/ home). Stocks were maintained on plates with nematode growth medium (NGM) seeded with HB101 *Escherichia coli* strain, under 12-h/12-h LD/CW cycle (400/0 lx and CW (18.5/20 °C, $\Delta = 1.5 \pm 0.125$ °C) environmental cycles. Transgenic animals were generated by microinjection of a P*sur-5::luc::gfp* construct at 50 or 100 ng/μL with the pRF4 marker (100 ng/μL)[26,65]. For bioluminescence recordings, transgenic P*sur-5::luc::gfp* nematode populations were synchronized to the same developmental stage by the chlorine method[66]. The harvested eggs were cultured overnight in a 50-mL Erlenmeyer flask with 3.5 mL of M9 buffer, 1× antibiotic-antimycotic (Thermo Fisher Scientific), and 10 μg/mL of tobramycin (Tobrabiotic, Denver Farma S.A., Buenos Aires, Argentina) at 110 rpm with a Vicking M23 shaker, in LD/CW (400/0 lx; 18.5/20 ° C, $\Delta = 1.5$ °C ± 0.125°C) conditions. The following day, L1 larvae were transferred to NGM plates at ZT1 and were grown for 48 h to the L4 stage under the same LD/CW cycle. Starting at ZT1 (1 h post lights on), the most fluorescent nematodes were selected manually under a SMZ100 stereomicroscope equipped with an epi-fluorescence attachment (Nikon, Minato, Tokyo) with a cool Multi-TK-LED light source (Tolket S.R.L., Buenos Aires, Argentina) to avoid warming the plate. Picking was performed in a room kept at a constant temperature (18 ° C). L4 larvae were selected after 48 h of light-dark/temperature entrainment and placed in a luminescence medium containing Leibovitz's L-15 medium without phenol red (Thermo Fisher Scientific) supplemented with 1× antibiotic–antimycotic (Thermo Fisher Scientific), 40 μM of 5-fluoro-2′-deoxyuridine (FUdR) to avoid new eclosions, 5 mg/mL cholesterol, 10 μg/mL tobramycin (Denver Farma S.A.), 1 mM D-luciferin (Gold Biotechnology, St Louis MO, USA), and 0.05% Triton X-100 to increase cuticle permeabilization. All chemical compounds were purchased from Sigma-Aldrich unless otherwise specified. Population nematode bioluminescence was recorded in 35-mm plate dishes (CELLSTAR, Greiner Bio-One, Kremsmünster, Austria) with 1 mL of the luminescence medium, by means of an AB-2550 Kronos Dio luminometer (Atto). The signal was integrated for 1 min, and readings were taken every 10 min.

For single-nematode measurements, P*sur-5::luc::gfp* nematodes at the L4 stage were selected as before, washed, and transferred directly to the liquid luminescence medium. A small, square piece of a transparent 96-well plate containing three × three wells was inserted inside a 35-mm dish plate to reduce the total volume and limit the nematode movement to the center of the plate. One nematode was placed in the center well with 200 μL of the luminescence medium. Plates were then transferred to the AB-2550 Kronos Dio luminometer. Single-nematode recordings were taken every 37 min with an integration time of 4 min.

In both cases, 2 μl of 10 mM DZNep (SML0305 in PBS 1×, pH 7.2) were added to the luminescence medium (200 μl/well), with a final concentration of 100 μM. Controls were treated with vehicle (PBS).

As previously described, custom MATLAB scripts (see code availability section below) were used to analyse the raw luminescence data[26]. This is necessary because of the high signal/noise ratio, especially for single worms. Briefly, data were trend-corrected by dividing by a fixed moving-average window of 24 h, then smoothed for 12 h. Data were further normalized to the maximum level of luminescence of all biological replicates along the entire time series on each experiment. After raw data clean-up by MATLAB, Period was estimated by BioDare2[61].

**Assay for effect of the inhibition of the methyl cycle on plant and algae circadian clock**. *Ostreococcus tauri* cells transgenically expressing a translational fusion of CCA1 to luciferase from the CCA1 promoter (CCA1-LUC)[30] were grown, imaged, and analyzed as described previously[49]. Briefly, cells were cultured under 12/12 h blue (Ocean Blue, Lee lighting filter 724) light/dark cycles (17.5 µE/m²/s) in Keller medium in artificial sea water. Cells were then transferred to 96-well microplates (Lumitrac, Greiner Bio-one), at a density of ~15 × 10⁶ cells/ml and entrained for 7–10 days before recording. One day before recording, 150 µl Keller medium in artificial sea water was replaced with 150 µl of Keller medium in artificial sea water containing 333 µM luciferin (Km+). Bioluminescence was measured on a TopCount (Packard) under constant darkness or constant red + blue LED light (5–12 µE/m²).

For protoplast isolation and luminescent imaging, plants expressing luciferase from the CCA1 promoter in the Col-0 background (kindly provided by Karen Halliday, University of Edinburgh) were grown in sterile soil under long-day conditions (16 h light/8 h dark) at 22 °C under 60–70 µmol/m²/s white LED tube lights (Impact T8). Protoplasts were isolated from 3-week old leaves by mechanically stripping the lower epidermis from the underlying mesophyll cells using magic tape, then releasing the mesophyll cells by incubating the remaining leaf tissue in a solution containing cellulase and pectinase R10 [67]. Protoplasts in a solution of 1 mM D-luciferin (Biosynth AG), 5% fetal bovine serum (Sigma-Aldrich), 50 µg/mL ampicillin, 140 mM NaCl, 115 mM CaCl2, 4.6 mM KCl, 1.86 mM MES pH 5.7 and 4.6 mM glucose were added to white, flat-bottomed 96-well plates (Lumitrac, Greiner Bio-one) at a concentration of 2 × 10⁵ cells/mL. DZnep (S7120, Selleck Chemicals LLC, Houston, TX, USA) was added to the protoplasts to achieve concentrations of 0, 0.125, 0.25, 0.5 or 1 µM from a 1 mM stock solution (prepared in distilled H₂O) to a total volume of 200 µL per well. Plates were sealed with a clear adhesive lid (TopSeal-A, Perkin Elmer). Protoplast luminescence was read by a LB942 Tristar[2] plate reader (Berthold Technologies GmbH & Co. KG, Bad Wildbad, Germany) every 50 min for 3 s per well, and kept under continuous red (630 nm) and blue (470 nm) LED light (5 µmol/m²/s each) at 19 °C.

*Chlamydomonas reinhardtii* strain CBR carrying a codon-adapted luciferase reporter driven by the *tufA* promoter in the chloroplast genome[68,69] was used. Culture preparation, bioluminescence monitoring, and data analysis were carried out as described previously[69]. Briefly, 5-day-old algal cultures on HS agar medium were cut out along with the agar by using a glass tube, and transferred to separate wells of a 96-well microtiter plates (Luciferin (final conc. 200 µM) and various concentrations of DZnep (Sigma-Aldrich) were added to the wells. Algae were synchronized by a single cycle of 12-h darkness/12-h light (30 µmol/m²/s) at 17 °C before bioluminescence monitoring using a custom-made luminometer-incubator[70] in DD at 17 °C. Period was estimated by BioDare2[61].

**Assay for effect of the inhibition of the methyl cycle on prokaryotic circadian clock**. *Vibrio harveyi* luciferase encoded by *luxA* and *luxB* (*luxAB*) genes was used as a luminescence reporter in cyanobacterium *Synechococcus elongatus* PCC 7942. The cyanobacterial clock-regulated reporter strain *kaiBCp::luxAB*[71] was selected for the test, expressing *luxAB* under the control of the promoter of the central clock genes *kaiBC*. *Synechococcus* strains were grown in modified BG11 media[72] supplemented with 40 µg/mL of spectinomycin. The cultures grown on BG11 agar plates for 3 days at 30 °C under continuous cool-white illumination (LL) (50 µE/m² s) were toothpicked onto fresh BG11 agar plates containing 0.015 g/L of L-methionine and different concentrations of 3-deazaneplanocin A hydrochloride (SML0305, Sigma-Aldrich) or sinefungin (Abcam, Cambridge, United Kingdom). After a single 12 h dark pulse for synchronization, *in vivo* luminescence rhythms were captured with a charge-coupled device (CCD) camera set up within a system to accommodate the agar plates containing the colonies[71]. Period was estimated by BioDare2[61].

**Measurement of *Hes7* oscillations in the mouse presomitic mesoderm**. Presomitic mesoderm (PSM) tissues from three littermate pHes7-UbLuc Tg embryos were embedded in 0.35% LMP-agarose/culture medium (10% FBS-DMEM/F12), in a silicon mold mounted onto a φ35mm-glass bottom dish, and luciferin-containing medium with DZnep (Sigma-Aldrich) or vehicle (MilliQ water) were added.

Time-lapse imaging of PSM was performed with an inverted microscope (IX81, Olympus, Tokyo, Japan) equipped with an Olympus x10 UPlanApo objective (N. A.: 0.8) and a VersArray cooled-CCD camera. 16-bit images were acquired every 5 min with Image-Pro Plus (Media Cybernetics, Inc., Rockville, MD, USA), with 2×2 and 4×4 binning and exposures of 100 ms and 4 m 25 s for DIC and chemiluminescent images, respectively. Raw imaging data were processed and quantified by ImageJ[73]. Period was estimated by BioDare2[61].

**Statistics and reproducibility**. Statistical analyses were conducted using GraphPad Prism 8 (GraphPad Software Inc., CA, USA). Sample size and the nature of replicates is specified in the corresponding sections. All experiments have been reproduced and results confirmed.

**Reporting summary**. Further information on research design is available in the Nature Research Reporting Summary linked to this article.

## Data availability

The source data underlying plots shown in main figures are provided in Supplementary Data 1. All other data shown in this work are available from J.M. Fustin upon reasonable request.

## Code availability

The Matlab code used for *C.elegans* rhythms analysis is available upon request to dgolombek@gmail.com or on Zenodo[74].

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

## Acknowledgements

This work was supported in part by the Ministry of Education, Culture, Sports, Science and Technology of Japan: Grant-in-aid for Scientific Research on Innovative Areas (26116713, J.-M. F.; 17H06401, H.K.), for Young Scientists (26870283, J.-M. F.), for Scientific Research A (15H01843, H.O.; 18H04015, H.O.), a grant for Core Research for Evolutional Science and Technology, Japan Science and Technology Agency (CREST/ JPMJCR14W3, H.O.). J.-M.F. was also supported by grants from the Kato Memorial Bioscience Foundation, the Senri Life Science Foundation (S-26003), the Mochida Memorial Foundation for Medical and Pharmaceutical Research, and the Kyoto University internal grant ISHIZUE, and H.O. was also supported by the Kobayashi International Scholarship Foundation. C.H.J. is supported by grants from the USA NIH/ NIGMS: GM067152 and GM107434. G.vO. is supported by a Royal Society University Research Fellowship (UF160685) and research grant (RGF\EA\180192). D.A.G. is supported by grants from the National Research Agency (ANPCyT, PICT-2015-0572) and the National University of Quilmes. E.G. was supported by the Wellcome Trust-University of Edinburgh Institutional Strategic Support Fund. M.V. and R.S. are funded by a grant from the Deutsche Forschungsgemeinschaft (STA421/7-1). J.-M. F. is a UKRI Future Leaders Fellow (MR/S031812/1). We thank Adrienne K. Mehalow, Jay C. Dunlap,

John O'Neill and Tokitaka Oyama for useful discussion and for kindly accepting to perform experiments. The authors thank G. Wolber, Freie Universität Berlin, Germany, for providing a LigandScout 4.2 license.

## Author contributions

Conceptualization, J.-M.F. and H.O.; Methodology, J.-M.F., M.V., C.R., S.J.C., T.K.T., Y.X., M.L.J., M.L.L., K.Y.-K., D.W., R.Ka., T.M., R.S., D.A.G., C.H.J., G.v.O., H.K.; investigation, J.-M.F., S.Y., C.R., M.Y., M.V., E.G., K.F., K.K., S.J.C., T.K.T., Y.X., M.L.J., R.Ko., M.L.L., K.Y.-K., T.M., D.A.G., G.v.O.; homology modeling, C.R.; original draft, J.-M.F.; reviewing & editing, J.-M.F., C.R., T.K.T, Y.X., K.Y.-K., D.W., R.Ka., T.M., R.S., D.A.G., C.H.J., H.K., G.v.O.; Funding Acquisition, J.-M.F., D.A.G., C.H.J., H.K., G.v.O and H.O.; Supervision, J.-M.F. Resources: S.T., L.H., H.K.

## Competing interests

The authors declare no competing interests.
