## [Peer Review File · Communications Biology]

Reviewers' comments:

Reviewer #1 (Remarks to the Author):

Fustin et al. describe an importance of methylation deficiency upon biological rhythms. Using cell lines, plants, algae, Cyanobacteria, and *C. elegans*, the authors examined biological changes of rhythms in the conditions of disrupted methylation cycles by means of 3-deazaneplanocin (DZnep), an inhibitor of adenosylhomocysteinase (AHCY), and the overexpression of MTAN. Although the manuscript is well written and the experiments technically sound and are well designed, there are five concerns to be addressed as follows.

Although it has been reported that 3-deazaadenosine (DAA) inhibiting AHCY affects m6A modification of mRNAs for *Per1*, *Per3*, and *Dbp* previously published in *Cell* (2013), is there any influence of DZnep on these genes?

Upon DZnep treatments, it is predictable that, accompanied with the accumulation of SAH, the pathway of homocysteine \rightarrow β -cystathionine \rightarrow Cys \rightarrow GSH may be repressed. Is there any influence of reduced levels of the pathway on the extension of rhythms by DZnep? Do the authors measure components of the pathway?

The authors measured the reduced levels of SAH. Do the authors check whether the overexpression of MTAN catalyzes the reaction of SAH \rightarrow S-ribosylhomocystein + adenine?

Are there any previous papers regarding rescue experiments of the MTAN overexpression in the mutants of Cyanobacteria *KaiC* or other organisms to change the biological rhythms?

It is known that S-ribosylhomocystein produced by MTAN is metabolized only in bacteria. Considering clinical approaches related to MTAN in the future, it may be a concern to some reduction of Cys and GSH. How do the authors think this point?

Reviewer #2 (Remarks to the Author):

This was an interesting and novel study demonstrating that biological rhythms are a quantitative gauge for methylation deficiency across the tree of life. They show that the AHCY inhibitor DZnep which disrupts the methyl cycle and increases SAH levels, increased period length of a rhythmic reporter gene in human, mouse, zebrafish, *Drosophila* cultures as well as mouse embryos, *Arapidosis* and algae. They then show that cyanobacteria are resistant to the effects of DZnep, hypothesizing a role for a secondary enzyme MTAN. In a convincing test of their MTAN hypothesis, they show that transfection of MTAN but not an enzymatically dead mutant MTAN protects both human and mouse cultures from the period-lengthening effect of DZnep. Together, these results importantly demonstrate that methyl deficiency leads to disrupted biological rhythms across highly divergent species and may have important translational implications for treating humans with AHCY mutations with MTAN from cyanobacteria.

Overall, the manuscript is well written and the methods and experimental conditions are all described clearly and rigorously.

The one major concern is that the authors do not know the main class of molecular targets of methylation regulating biological rhythms. They hypothesize that RNA methylation is responsible, but

do not show this. While I understand the difficulty of investigating all potential targets across all species, the manuscript would be much stronger if they could test their hypothesis that RNA methylation is significantly altered using the mammalian cell conditions of Figure 5.

Reviewer #3 (Remarks to the Author):

The manuscript by Justin et al. provides a comprehensive description of the potential effects of methylation state changes on biological rhythms, proposing this is as universal mechanism from bacteria, algae, plants, invertebrates to mammals. The authors center the analysis on AHCY, the enzyme mediating a crucial step on the one-carbon metabolism, converting SAH to homocysteine. In silicon analysis of protein structure showed high conservation across different taxa and is followed by in vitro assays showing a dose response to DzNEP, an inhibitor of AHCY, which affect circadian rhythmicity increasing period length. Lastly, expression of bacterial SAH nucleosidase MTAN in mouse and human cells is shown to rescue the effects of Dznep- induced methyl potential decay on the clock. While the work is important in our understanding of global methylation mechanisms that have a broad impact in physiology and human health, is limited by a rather comparative and descriptive approach lacking mechanistic insights; particularly after excellent previous work by the group showing the role of RNA methylation in the circadian clock.

Specific comments:

1-Fig 2 is missing a quantification of SAH after exposure to Dznep in the cells tested. This should be consistent across figures and is important to demonstrate the effects on SAH are present at the tested concentrations.

2-Homocysteine is an inhibitor for DNA-methyl transferases (DNMT1-3a-3b). What is the effect of Dznep in their activities? what is the effect on global methylation? This aspect of the methylation cycle is completely omitted in the study, despite previous work in mouse and human cells shown potent effects of DNMT inhibitors on the circadian machinery (Azzi et al PMID24531307 and Croci et al PMID 27883893).

3-While the idea in the paper is the circadian clock being a "quantitative gauge for methylation deficiency" there is no explanation as what is the mechanism sensing. Is this a direct effect of methylation in clock core components as proposed in previous work?

4-Histone methylation can directly affect the clock: for example, the rhythms of H3K4me3 at the promoter of Dbp and Per1 enables cycles of gene expression , mediated by binding of MLL1 to Bmal1:Clock dimers. It is important to understand whether the effects for Dznep and MTAN are mediated by changes in methylation at this level.

5-Global methylation, and preferably, gene specific methylation should be profiled in the experiments expressing MTAN, to understand the potential off target effects of broadly affecting methylation and also to uncover the mechanisms that mediate this effect.

6-Minor comments: Labels and axis in inserts in Figures 3 and 5 need to be revised for repetitions and naming variables analyzed by ANOVA that don't appear in the graph.

Reviewer #1 (Remarks to the Author):

Fustin et al. describe an importance of methylation deficiency upon biological rhythms. Using cell lines, plants, algae, Cyanobacteria, and *C. elegans*, the authors examined biological changes of rhythms in the conditions of disrupted methylation cycles by means of 3-deazaneplanocin (DZnep), an inhibitor of adenosylhomocysteinase (AHCY), and the overexpression of MTAN. Although the manuscript is well written and the experiments technically sound and are well designed, there are five concerns to be addressed as follows.

Thank you very much for this positive assessment. Please find our replies below.

Although it has been reported that 3-deazaadenosine (DAA) inhibiting AHCY affects m6A modification of mRNAs for *Per1*, *Per3*, and *Dbp* previously published in *Cell* (2013), is there any influence of DZnep on these genes?

Per1, *Per3* and *Dbp* are mammalian clock genes, mostly not conserved in invertebrates, plants, algae or bacteria. Our manuscript is about the evolutionary conservation of the link between the methyl cycle and the circadian clock. We do not think providing data on the effect of DZnep on mammalian-specific clock genes would provide any insights on what conserved mechanisms link the methyl cycle with the clock. However, in response to this comment and to reviewer #3, we now show that indeed DZnep does reduce global mRNA m6A in mammalian cells, which is rescued by WT MTAN (Fig. 5e)

Upon DZnep treatments, it is predictable that, accompanied with the accumulation of SAH, the pathway of homocysteine \rightarrow β -cystathionine \rightarrow Cys \rightarrow GSH may be repressed. Is there any influence of reduced levels of the pathway on the extension of rhythms by DZnep? Do the authors measure components of the pathway?

It is unlikely that GSH, synthesized from cysteine, will be strongly affected because of the amount of free cysteine in the medium. However, to address this point, we quantified homocysteine, cystathionine, cysteine, GSH and GSSH in the same samples used for Fig. 5c.

We added the following paragraphs in the discussion (see also related new method section in the manuscript):

That mammalian cells expressing *E. coli* MTAN are completely protected from the period-lengthening effects of DZnep demonstrates that these effects are solely dependent upon methyl cycle inhibition and SAH accumulation. Moreover, we have shown in mouse cells that the functional consequences of methyl cycle inhibition, *i.e.* a decrease in the methylation of proteins and mRNA, is also rescued by WT MTAN. This confirms that MTAN, despite removing SAH from the cycle in a way that does not normally occur in mammalian cells, allows the methyl cycle to turn and drive cellular methylations when AHCY does not function. MTAN hydrolyses SAH to S-ribosylhomocysteine and adenine, not to Hcy and adenosine as AHCY does. Since Hcy is needed for the regeneration of methionine in the methyl cycle, how could the methyl cycle still maintain cellular methylations active without Hcy? It is likely that, with the abundance of free methionine in the cell, the synthesis of SAM and downstream methylations can continue unimpaired even when Hcy is low. Hcy is however also used for the synthesis of glutathione (GSH) via cystathionine and cysteine, a metabolic route called the transsulfuration pathway (Hcy \rightarrow cystathionine \rightarrow cysteine \rightarrow GSH), which could potentially have been inhibited by MTAN expression and DZnep. To investigate potential disruption of this pathway caused by either DZnep or MTAN, we quantified these metabolites from cell extracts used in Fig. 5c. As expected, expression of WT MTAN caused a significant decrease in Hcy compared to cells expressing D197A (Fig. S6). Overall, however, the transsulfuration pathway appeared more active in cells expressing WT MTAN, especially given the significantly higher concentrations of cysteine and GSH, which were probably due to the significant increase in SAM (Fig. S5c), a powerful allosteric activator of this pathway⁴⁶. A general tendency of DZnep to inhibit the transsulfuration pathway was observed, more pronounced in cells expressing

the mutant MTAN. Glutathione disulfide (GSSG), generated during enzyme-catalyzed reduction reactions of GSH with peroxides or disulfide bonds, tended to decrease under DZnep treatment. The GSH/GSSH ratio, a measure of oxidative stress, was not significantly affected by either DZnep or MTAN, indicating that the mild effects of DZnep and MTAN expression on the transsulfuration pathway are unlikely to contribute to the circadian period lengthening. These data do show that WT MTAN also rescues the cells from the mild inhibition of the transsulfuration pathway by DZnep, however.

The authors measured the reduced levels of SAH. Do the authors check whether the overexpression of MTAN catalyzes the reaction of SAH -> S-ribosylhomocystein + adenine?

In Figure 5c we did show that expression of WT MTAN, not D197A mutant, prevents the increase of SAH in the cells treated with DZnep. It stands to reason that if WT MTAN prevents the increase of SAH, but not the D197A enzyme dead mutant, it is because of the activity of the MTAN enzyme. The decrease of SAH is what we want to achieve, and the increase in adenine and ribosylhomocysteine is secondary. The main point, that SAH accumulation is prevented by MTAN expression, has been clearly demonstrated. Furthermore, there is no guarantee that adenine and ribosylhomocysteine will increase, as it will depend on what other enzymatic activities working in the cell can further metabolize these metabolites, especially adenine, which is readily salvaged and degraded when it is in excess.

However, we now confirm the expression of WT and D197A MTAN in mouse and human cells (Fig. S5a, b).

Are there any previous papers regarding rescue experiments of the MTAN overexpression in the mutants of Cyanobacteria KaiC or other organisms to change the biological rhythms?

To our knowledge this has never been done, in any organisms, and is a very interesting suggestion!

It is known that S-ribosylhomocystein produced by MTAN is metabolized only in bacteria. Considering clinical approaches related to MTAN in the future, it may be a concern to some reduction of Cys and GSH. How do the authors think this point?

Thank you for this comment. It is true that the removal of SAH by MTAN may lead to a decrease in the transsulfuration pathway. If this occurred, co-treatment with cysteine may provide the solution. However, cysteine is an amino acid abundant in eggs, chicken, meat and grains; cysteine (and GSH) deficiency is therefore unlikely even when MTAN is expressed. Moreover, we reply to this comment by providing new data on homocysteine, cystathionine, cysteine, GSH and GSSH and showing no reduction in Cys and GSH with WT MTAN (Fig. S6). Please also refer to our answer to your second comment, and see the new text in the methods related to these new data.

Reviewer #2 (Remarks to the Author):

This was an interesting and novel study demonstrating that biological rhythms are a quantitative gauge for methylation deficiency across the tree of life. They show that the AHCY inhibitor DZnep which disrupts the methyl cycle and increases SAH levels, increased period length of a rhythmic reporter gene in human, mouse, zebrafish, *Drosophila* cultures as well as mouse embryos, *Arapidosis* and algae. They then show that cyanobacteria are resistant to the effects of DZnep, hypothesizing a role for a secondary enzyme MTAN. In a convincing test of their MTAN hypothesis, they show that transfection of MTAN but not an enzymatically dead mutant MTAN protects both human and mouse cultures from the period-lengthening effect of DZnep. Together, these results importantly demonstrate that methyl deficiency leads to disrupted biological rhythms across highly divergent species and may have important translational implications for treating humans with AHCY mutations with MTAN from cyanobacteria.

Overall, the manuscript is well written and the methods and experimental conditions are all described clearly and rigorously.

Thank you very much for this positive review. Please find our replies below.

The one major concern is that the authors do not know the main class of molecular targets of methylation regulating biological rhythms. They hypothesize that RNA methylation is responsible, but do not show this. While I understand the difficulty of investigating all potential targets across all species, the manuscript would be much stronger if they could test their hypothesis that RNA methylation is significantly altered using the mammalian cell conditions of Figure 5.

Thank you very much for this comment. We have now screened the most likely candidates (lysine and arginine methylation, *N*⁶-methyladenosine in total RNA and mRNA, and 5-methylcytosine in DNA) and reveal that, *N*⁶-methyladenosine in mRNA is significantly inhibited by DZnep and rescued by WT but not by D197A MTAN (Fig 5e). We also show that lysine and arginine methylation of some proteins, but not all, is likewise inhibited by DZnep and rescued by WT MTAN (Fig. 5d and S5e), while *N*⁶-methyladenosine in total RNA (Fig. S5f) and 5-methylcytosine in DNA (Fig. S5g) are not significantly affected.

We also added the following text in the results:

Finally, we sought to determine the effects of DZnep on global lysine and arginine methylation, on RNA and mRNA *N*⁶-methyladenosine (m6A), and on DNA 5-methylcytosine (m5C), as well as the rescue of their potential inhibition by MTAN in mouse cells. In all cases, PER2::LUC MEFs were treated with DZnep 0, 5 or 10 \$\mu\$ M (to obtain near-maximum period lengthening) for 48-hours.

Immunoblotting using an antibody against mono- and di- methylated lysine showed that a few proteins had lower methylated levels under DZnep treatment, but only in cells transfected with the mutant inactive MTAN (Fig. 5d). Probing mono- and di-methylated arginine in the same cells showed somewhat less pronounced methylation inhibition, but was also rescued by WT MTAN (Fig. S5e). We also checked the global levels of the specific methyl marks Histone 4 Arginine 3 symmetric demethylation (transcriptional repression) and Histone H3 Lysine 4 trimethylation (transcriptional activation) but little inhibition was seen (Fig. S5e).

N⁶-methyladenosine did not significantly fluctuate between treatments when measured from total RNA samples, composed mainly (>90%) of 18S and 28S rRNA, each macromolecule containing only one single m6A site (Fig. S5f).

When quantified from mRNA, however, m6A showed a significant decrease under DZnep treatment, but only in cells transfected by the mutant inactive MTAN (Fig. 5e). In contrast, DNA m5C was not significantly affected (Fig S5g).

It is possible that a longer treatment with DZnep may have significantly affected more stable methylations such as m6A in rRNA and m5C in DNA or caused a more widespread inhibition of histone methylation, but since the period lengthening effects of DZnep are observable by 24 hours we speculate these methylations may not contribute to the period lengthening. Promoter-specific DNA or histone methylation, for example in the promoter of clock genes, which would not be detectable when global methylation is quantified, may also be inhibited by DZnep.

In conclusion, we have shown in mammalian cells that the period lengthening effects of DZnep exclusively depends on the inhibition of AHCY and the increase in SAH, mainly leading to protein and mRNA m6A methylation inhibition. More significantly, the partial rewiring of the mammalian methyl cycle, protecting proteins and nucleic acids from the consequences of methyl cycle inhibition, might present a conceptually novel opportunity to treat methylation deficiencies, such as homocysteinemia or the autosomal recessive AHCY deficiency^{12, 38-41}.

Reviewer #3 (Remarks to the Author):

The manuscript by Justin et al. provides a comprehensive description of the potential effects of methylation state changes on biological rhythms, proposing this as a universal mechanism from bacteria, algae, plants, invertebrates to mammals. The authors center the analysis on AHCY, the enzyme mediating a crucial step on the one-carbon metabolism, converting SAH to homocysteine. In silicon analysis of protein structure showed high conservation across different taxa and is followed by in vitro assays showing a dose response to DzNEP, an inhibitor of AHCY, which affect circadian rhythmicity increasing period length. Lastly, expression of bacterial SAH nucleosidase MTAN in mouse and human cells is shown to rescue the effects of Dznep-induced methyl potential decay on the clock.

While the work is important in our understanding of global methylation mechanisms that have a broad impact in physiology and human health, is limited by a rather comparative and descriptive approach lacking mechanistic insights; particularly after excellent previous work by the group showing the role of RNA methylation in the circadian clock.

Thank you very much for the very careful reading of our manuscript. However, please note the first author of this manuscript is Fustin, not Justin. Please find our replies below.

Specific comments:

1-Fig 2 is missing a quantification of SAH after exposure to DZnep in the cells tested. This should be consistent across figures and is important to demonstrate the effects on SAH are present at the tested concentrations.

We agree that, if only for the sake of thoroughness, SAM/SAH quantification should be done in zebrafish, drosophila, mouse embryos and human cells. However, we selected mouse cells as a representative animal from Fig 2, to be compared with the evolutionarily most distant eukaryotes (unicellular algae) from Fig. 3. Due the virtually identical response of the clock in the other eukaryotes tested, and to the homology modelling shown in Fig. 1, we consider it highly likely that DZnep will have the same effect (AHCY inhibition, hence SAH accumulation) in other organisms. Moreover, given the collaborative and intercontinental nature of our manuscript and the current circumstances of our co-authors, such an experiment could not be easily completed.

2-Homocysteine is an inhibitor for DNA-methyl transferases (DNMT1-3a-3b). What is the effect of Dznep in their activities? what is the effect on global methylation? This aspect of the methylation cycle is completely omitted in the study, despite previous work in mouse and human cells shown potent effects of DNMT inhibitors on the circadian machinery (Azzi et al PMID24531307 and Croci et al PMID 27883893).

Apologies for contradicting this reviewer, but homocysteine is not an inhibitor of DNA methylation. High levels of homocysteine are associated with high levels of SAH, partly because excess of homocysteine causes AHCY to work in reverse and synthesize SAH (de la Haba and Cantoni, 1958), which is the actual inhibitor of DNA methylation (see for example Jamaluddin et al., 2007).

Moreover, DNA methylation is not a component of the methyl cycle. DNA methylation is only one of the outputs of the methyl cycle, and DNA methyltransferases are only 3 out of 600 methyltransferases identified so far in human, many of functions unknown. Our manuscript is no about DNA methylation, and we see no reason to focus on DNA methylation in the current manuscript.

Nevertheless, we now provide data showing that 48 hours of DZnep treatment (5 and 10 microM) in mouse cells do not significantly affect 5-methylcytosine in DNA, while lysine and arginine methylation of some proteins, as well as *N*⁶-methyladenosine in mRNA, but not total RNA, are significantly inhibited by DZnep and rescued by WT, but not D197A, MTAN (Fig 5d,e; S5e, f, g).

We also added the following text in the manuscript:

Finally, we sought to determine the effects of DZnep on global lysine and arginine methylation, on RNA and mRNA *N*⁶-methyladenosine (m6A), and on DNA 5-methylcytosine (m5C), as well as the rescue of their potential inhibition by MTAN in mouse cells. In all cases, PER2::LUC MEFs were treated with DZnep 0, 5 or 10 μM (to obtain near-maximum period lengthening) for 48-hours.

Immunoblotting using an antibody against mono- and di- methylated lysine showed that a few proteins had lower methylated levels under DZnep treatment, but only in cells transfected with the mutant inactive MTAN (Fig. 5d). Probing mono- and di-methylated arginine in the same cells showed somewhat less pronounced methylation inhibition, but was also rescued by WT MTAN (Fig. S5e). We also checked the global levels of the specific methyl marks Histone 4 Arginine 3 symmetric demethylation (transcriptional repression) and Histone H3 Lysine 4 trimethylation (transcriptional activation) but little inhibition was seen (Fig. S5e).

*N*⁶-methyladenosine did not significantly fluctuate between treatments when measured from total RNA samples, composed mainly (>90%) of 18S and 28S rRNA, each macromolecule containing only one single m6A site (Fig. S5f).

When quantified from mRNA, however, m6A showed a significant decrease under DZnep treatment, but only in cells transfected by the mutant inactive MTAN (Fig. 5e). In contrast, DNA m5C was not significantly affected (Fig S5g).

It is possible that a longer treatment with DZnep may have significantly affected more stable methylations such as m6A in rRNA and m5C in DNA or caused a more widespread inhibition of histone methylation, but since the period lengthening effects of DZnep are observable by 24 hours we speculate these methylations may not contribute to the period lengthening. Promoter-specific DNA or histone methylation, for example in the promoter of clock genes, which would not be detectable when global methylation is quantified, may also be inhibited by DZnep.

In conclusion, we have shown in mammalian cells that the period lengthening effects of DZnep exclusively depends on the inhibition of AHCY and the increase in SAH, mainly leading to protein and mRNA m6A methylation inhibition. More significantly, the partial rewiring of the mammalian methyl cycle, protecting proteins and nucleic acids from the consequences of methyl cycle inhibition, might present a conceptually novel opportunity to treat methylation deficiencies, such as homocysteinemia or the autosomal recessive AHCY deficiency^{12, 38-41}.

3-While the idea in the paper is the circadian clock being a "quantitative gauge for methylation deficiency" there is no explanation as what is the mechanism sensing. Is this a direct effect of methylation in clock core components as proposed in previous work?

Yes, it is possible that some clock components (clock genes promoters, mRNAs, proteins), not necessarily the same in different species, may directly link the clock with the methyl cycle. However, we favor a less "clock specific" mechanism, and rather think the methyl cycle as a metabolic sensor that can regulate protein output in an acute (mRNA or protein methylation) or chronic (DNA or histone methylation) way in relation to the metabolic state of the organisms. Circadian rhythms, because they are easy to quantify at the molecular and behavioral levels, are a good way to study how the methyl cycle operates such regulation. How the methyl cycle exactly regulates the clock in different organisms is unknown but is likely to be via a combination of RNA and protein methylation.

4-Histone methylation can directly affect the clock: for example, the rhythms of H3K4me3 at the promoter of *Dbp* and *Per1* enables cycles of gene expression, mediated by binding of MLL1 to Bmal1:Clock dimers. It is important to understand whether the effects for Dznep and MTAN are mediated by changes in methylation at this level.

No, it is not really important for the scope of the current manuscript because *Dbp*, *Per1*, *Bmal1* and *Clock* are mammalian clock genes that are conserved in only a few species investigated here, if any, while the theme of our paper is the conservation of the link between the methyl cycle and the clock. The experiments with MTAN here provide clear evidence that the mammalian methyl cycle can be rewired by bacterial enzymes, making mammalian cells resistant to AHCY inhibition.

As previously mentioned in our response to comment 2, we now show the effects of a 48-hours treatment of DZnep in cells expressing WT or D197A MTAN are likely mediated by global changes at the level of N^6 -methyladenosine in mRNA, while only a few unidentified proteins show parallel changes in lysine and arginine methylation. In contrast, N^6 -methyladenosine in total RNA and 5-methylcytosine in DNA appear not affected by such a treatment.

We however now mention this in the results section:

It is possible that a longer treatment with DZnep may have significantly affected more stable methylations such as m6A in rRNA and m5C in DNA or caused a more widespread inhibition of histone methylation, but since the period lengthening effects of DZnep are observable by 24 hours we speculate these methylations may not contribute to the period lengthening. Promoter-specific DNA or histone methylation, for example in the promoter of clock genes, which would not be detectable when global methylation is quantified, may also be inhibited by DZnep.

5-Global methylation, and preferably, gene specific methylation should be profiled in the experiments expressing MTAN, to understand the potential off target effects of broadly affecting methylation and also to uncover the mechanisms that mediate this effect.

We assume that this reviewer —again— refers to DNA and/or histone methylation? If it is about DNA methylation, see our response to comment 2. If this is about histone methylation, see our response to comment 4.

6-Minor comments: Labels and axis in inserts in Figures 3 and 5 need to be revised for repetitions and naming variables analyzed by ANOVA that don't appear in the graph.

Figures have been thoroughly checked and no mistakes or missing information were spotted.

REVIEWERS' COMMENTS:

Reviewer #1 (Remarks to the Author):

The authors made honest efforts to respond to the reviewer's critiques. In particular, they provide new experiment data and explain what they add and how they change. The referee agrees with their revisions.

Reviewer #2 (Remarks to the Author):

The authors have appropriately responded to the prior critiques.

Reviewer #3 (Remarks to the Author):

All comments addressed